# Learning the diffusion of nanoparticles in liquid phase TEM via physics-informed generative AI

Zain Shabeeb ⓘ , Naisargi Goyal, Pagnaa Attah Nantogmah & Vida Jamali ⓘ ✉

The motion of nanoparticles in complex environments can provide us with a detailed understanding of interactions occurring at the molecular level. Liquid phase transmission electron microscopy (LPTEM) enables us to probe and capture the dynamic motion of nanoparticles directly in their native liquid environment, offering real time insights into nanoscale motion and interaction. However, linking motion to interactions to decode the underlying mechanisms of motion and interpret interactive forces at play is challenging, particularly when closed-form Langevin-based equations are not available to model the motion. Herein, we present LEONARDO, a deep generative model that leverages a physics-informed loss function and an attention-based transformer architecture to learn the stochastic motion of nanoparticles in LPTEM. We demonstrate that LEONARDO successfully captures statistical properties suggestive of the heterogeneity and viscoelasticity of the liquid cell environment surrounding the nanoparticles.

Nature operates under the influence of stochasticity. This stochasticity, which shows up in the motion of nanoparticles, e.g., quantum dots in polymer matrices[1], proteins on lipid membranes[2–4], and vesicles in cells[5], is closely related to the nanoparticles' interactions with the complex environments surrounding them. Probing such nanoscale motion and interaction is challenging using conventional microscopy techniques due to the nanoscale, spatially heterogeneous, and dynamic nature of interactive forces, which require spatiotemporal resolution beyond what these methods can offer. In situ liquid phase transmission electron microscopy (LPTEM) is a new microscopy method that enables direct imaging of the motion of nanoparticles in their native liquid environment inside a microfluidic liquid cell chamber and using a transmission electron microscope. The motion of nanoparticles in a liquid and in interaction with the window membrane of the LPTEM microfluidic chamber is stochastic and complex (Fig. 1a)[6–14]. The lack of a computational model to capture the complex motion and interaction of nanoparticles in LPTEM has limited the application of LPTEM for single-particle imaging. This is despite the unprecedented spatial and temporal resolution (nanometer and millisecond) that LPTEM offers for resolving nanoscale motion and interaction[15–17].

Several ideal stochastic processes with closed-form equations exist that can describe particles' motion in certain types of environments. For example, Brownian motion[18] captures the non-correlated random displacements of a particle's trajectory due to the presence of thermal noise, fractional Brownian motion (FBM)[19] describes the short- and long-range correlations of the displacements of a trajectory moving in a highly viscoelastic environment, and continuous time random walk (CTRW)[20] models the trapping and escaping events of a particle as it moves over a random energy landscape with potential wells of various depths. In many experimental systems, including LPTEM, motion is a complex and hybrid mixture of various stochastic processes in space and time[6,21–26]. This complex motion is due to the nature of interactions that encompass different types of forces, which are nanoscopic and heterogeneous, that is, spatially varying and often hierarchical. The mathematical framework of the generalized Langevin equation that relates the position of particles to the stochastic noise present in the system through the fluctuation-dissipation theorem can potentially model more complex types of stochastic motion[27,28]. However, implementing this framework requires specifying the type of noise and the memory kernel term describing the past history of the system[29]. Computing the memory kernel in complex environments,

School of Chemical and Biomolecular Engineering, Georgia Institute of Technology, 311 Ferst Drive NW, Atlanta, GA 30332, USA. ✉e-mail: vida@gatech.edu

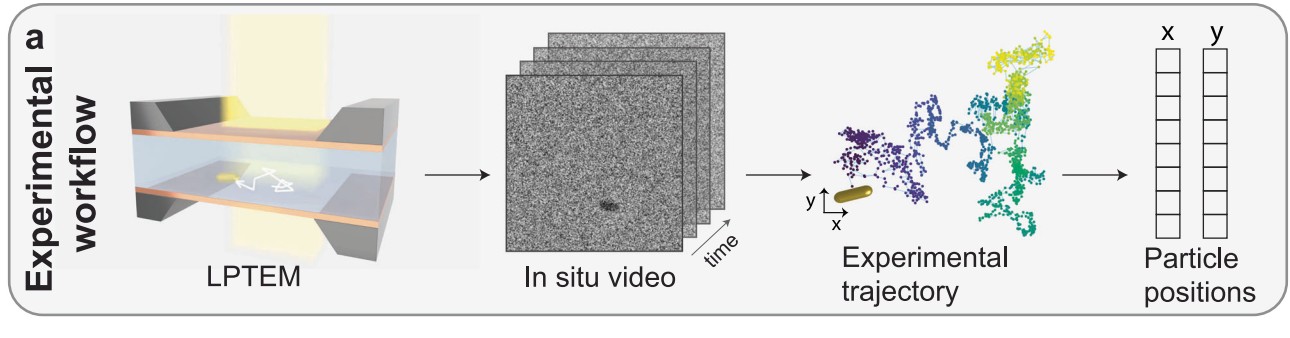

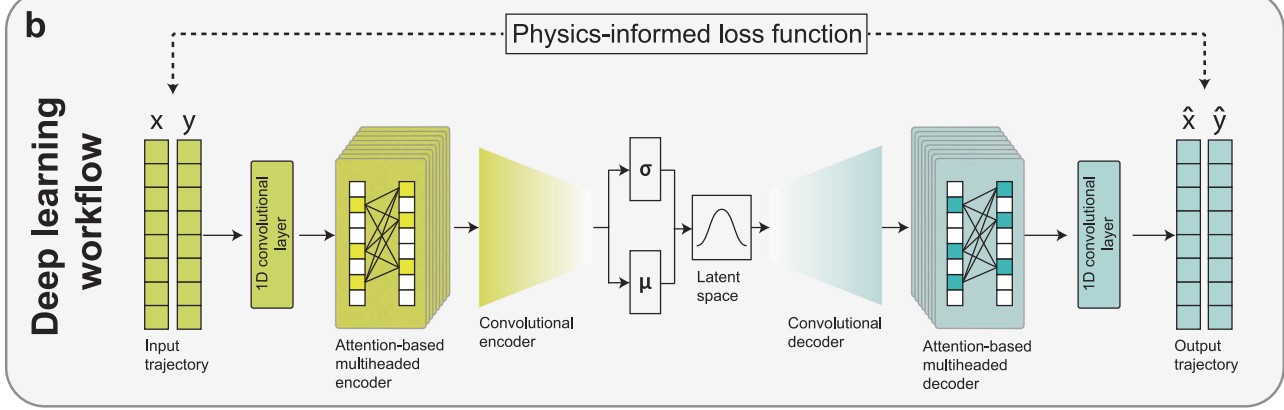

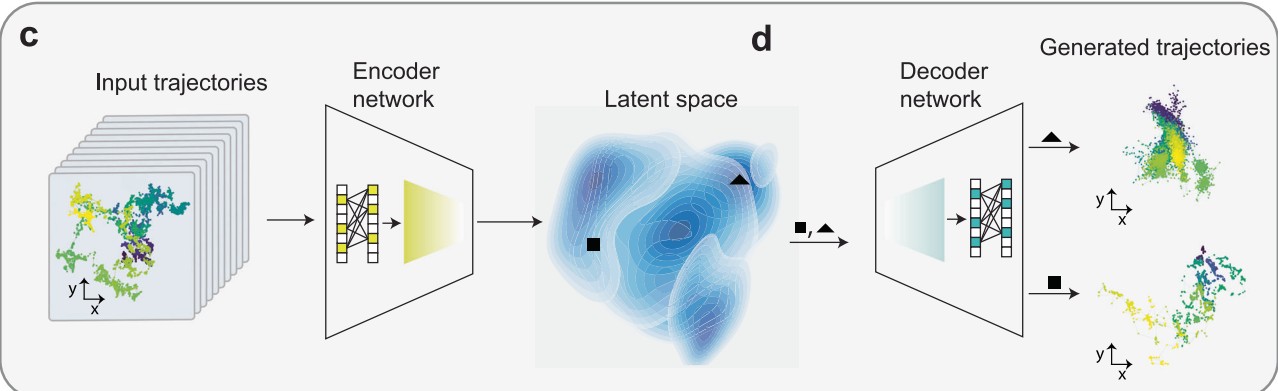

**Fig. 1 | Overview of LEONARDO workflow. a** Schematic overview of our workflow for extracting single particle trajectories from LPTEM movies. By imaging the stochastic motion of single gold nanorods in water as they move and interact with the window of the liquid cell microfluidic chamber of LPTEM, we collect a large dataset of single-particle trajectories from LPTEM experiments. **b** Schematic of the LEONARDO model, a transformer-VAE with self-attention mechanisms in the encoder and decoder, mapping input trajectories to a low-dimensional latent space and reconstructing them with the aid of a physics-informed loss function. **c** Latent space representation of unseen trajectories encoded by the trained model, showing distinct clustering and overlaps indicative of different diffusion behaviors. **d** Demonstration of the generative power of LEONARDO in simulating new synthetic LPTEM trajectories by sampling from different regions of the latent space.

including those with heterogeneous energy landscapes, using approaches such as molecular dynamics simulations is extremely difficult and often requires assumptions about the functional form of the memory kernel[30–32]. Therefore, obtaining a mechanistic understanding and modeling of the motion of particles in environments with heterogeneous interaction energy landscapes have been elusive at the nanoscale.

Recent advancements in artificial intelligence (AI) have led to the development of supervised machine learning methods that correlate experimental observations with different classes of ideal stochastic processes[33,34]. For example, supervised deep convolutional neural networks trained on simulated ideal stochastic processes have been used to classify the underlying mechanism of motion associated with single nanoparticle trajectories from LPTEM experiments[6]. Other studies have also used similar supervised learning methods with different neural network architectures to classify experimental single-particle trajectories into categories of known stochastic processes[35–39]. Unsupervised learning methods have also been employed to characterize the underlying mechanism of motion by training deep learning models on simulated data. For example, it was shown that standard autoencoders can be used to reconstruct ideal stochastic diffusion processes and identify their physical parameters[40]. Additionally, variational autoencoders (VAEs) with a probabilistic decoder were used to model ideal stochastic diffusion for Gaussian processes[41]. However, capturing the wide array of Gaussian and non-Gaussian diffusion characteristics exhibited by particles in experimental data is not achievable through models trained solely on ideal stochastic processes.

The rise of generative AI has provided new approaches to developing data-driven models that learn hidden features of training data and creating black-box simulators with the ability to generate entirely new synthetic data. For example, deep generative models such as VAEs

have proven to be effective in learning low-dimensional latent space representations of complex data in different domains and generating synthetic data[42]. VAEs achieve this task by mapping high-dimensional data to a lower-dimensional probabilistic latent space through a process of encoding, sampling, and decoding, thereby capturing the essential characteristics of the input data[42]. Through this process, the model learns latent variables that each follow a prior distribution (e.g., Gaussian) that serve as the parameters of the black-box simulation model.

Herein, we pose the question: can generative AI learn and model the complex surface diffusion of nanoparticles from LPTEM experiments? If so, how can that be achieved? To answer these questions, we introduce a VAE deep learning model, named LEONARDO, with an attention-based transformer architecture[43], and a customized physics-informed loss function to learn the complex diffusion of nanoparticles in LPTEM from tens of thousands of short single-particle trajectories (Fig. 1b). The self-attention mechanism captures complex temporal dependencies in time series data and has proven to be effective in a wide range of applications, such as the grammar of natural languages[44,45], the intricate molecular structures of chemical species[46], and the complex patterns of musical sequences[47]. The physics-informed loss function is designed to quantify the deviations between key statistical features of the input and generated trajectories. These features, such as the moments of the distribution, are canonically used in defining and characterizing stochastic diffusion processes, and therefore, aid the model in learning important attributes of trajectories. Leveraging the power of the self-attention mechanism, LEONARDO learns the temporal dependencies within single nanoparticle trajectories from LPTEM. We demonstrate that LEONARDO captures key physical properties related to the interaction energy landscape, including the non-Gaussianity of the distribution of displacements of a trajectory and their temporal correlations, which relate to the heterogeneity of potential wells in the environment surrounding the particle and their caging effect due to viscoelasticity. The generative power of LEONARDO enables us to use it as a black-box simulator by generating synthetic trajectories that are similar to real experimental data and cover the entire observed regime of behavior at different electron beam dose rates of the microscope and particle size (Fig. 1c, d). This feature provides an opportunity to generate ample particle trajectory data for downstream tasks, particularly in automating electron microscopes, where large and diverse datasets are essential for training algorithms that control the microscope[48].

## Results and discussion
### Imaging and learning single nanoparticle trajectories from LPTEM experiments using the LEONARDO framework
To learn the stochastic motion of the nanoparticles in LPTEM, we used a model system of gold nanorods diffusing in water. We first curated a large dataset of single-particle trajectories of gold nanorods dispersed in water (see Methods for sample preparation details) and moving near the silicon nitride ($SiN_x$) membrane window of the liquid cell chamber of LPTEM. A single nanoparticle trajectory from LPTEM, although highly stochastic in nature, is essentially a sequence of steps that the nanoparticle takes in time, containing all embedded time dependencies and, therefore, the information on its interactions with the surrounding environment over time. Figure 1a illustrates the workflow used here to collect single nanoparticle trajectories from LPTEM experiments. In situ movies of the stochastic motion of gold nanorods in LPTEM were recorded in real time. The collected movies were processed and analyzed as described in the Methods section to arrive at a large dataset of single nanoparticle trajectories (Fig. 1a). The normalized $x$ and $y$ coordinates of the two-dimensional (2-D) trajectories (see Supplementary Information for the normalization procedure) were used to formulate the training data for the neural network model with the architecture illustrated in Fig. 1b and Supplementary Fig. S3.

To generalize the model to all experimental conditions applicable to LPTEM experiments, a diverse training dataset was prepared by collecting 38,279 short (200-frame) experimental trajectories at different camera frame rates and a broad range of electron beam dose rates of the microscope (2 to 60 $e^-/Å^2 \cdot s$) for different sizes of gold nanorods (20 to 60 nanometers (nm) long). The trajectory length of 200 frames is a balanced choice to capture sufficient particle dynamics and accommodate datasets of varying sizes collected across different video frame rates. Longer experimental trajectories can be segmented into shorter fixed-length pieces to capture local dynamics, while subsampling of long trajectories can be used to study longer-range statistics with LEONARDO.

The input trajectories, after passing through a convolutional layer to increase the embedding dimension from 1 to 128, were transformed in the encoder network that features two sequential multiheaded self-attention blocks inspired by the transformer architecture[43] to capture the temporal dependencies inherent in the trajectories (see the Supplementary Information and Methods section for details). These blocks feed into a convolutional encoder that compresses the output into a 12-dimensional latent vector, $\mathbf{z}$, where each dimension follows a prior standard Gaussian distribution, with the help of a relative-entropy loss function term[42]. The latent vector is subsequently expanded back to the original trajectory length after passing through the decoder and the final convolutional layer (Fig. 1b). Through this transforming, compressing, and expanding process, the model learns to encode the input data into 12 independent Gaussian distributions (with mean $\boldsymbol{\mu}$ and standard deviation $\boldsymbol{\sigma}$, Fig. 1b) and subsequently samples from those distributions are used to reconstruct the input data in the output.

**Physics-informed loss function.** Given the stochastic nature of the trajectories, it is impractical to aim for an exact reconstruction after passing through a low-dimensional latent space using standard loss functions such as mean-squared error (MSE) and relative-entropy loss, as previously noted in the literature[41]. Therefore, the aim of this work is not to reconstruct the trajectories exactly; rather, it is to reconstruct their statistical properties and to learn the underlying physics related to those statistical properties. Thus, we designed a physics-informed loss function that includes terms customized to learn stochastic trajectories in addition to the standard loss terms in a VAE and we assigned a lower weight to the standard MSE-based reconstruction term to minimize its contribution to the total loss (see Supplementary Information for details of each loss term). Physics-informed machine learning has recently gained traction in applications where partial knowledge of the underlying physics of the input data is incorporated into the model's loss function[49]. This approach constrains the learning process with physical laws, ensuring that the model captures the underlying physics of the system.

The new loss function proposed here includes additional terms comprised of errors between statistical moments of the distribution of displacements, $\Psi(\delta\mathbf{r}(t))$, where $\delta\mathbf{r}(t) = \mathbf{r}(t) - \mathbf{r}(t-1)$ is the nanoparticles' displacement between two consecutive frames, and $\mathbf{r}(t) = (x(t), y(t))$ is the position vector of the nanoparticle at time $t$, with $x(t)$ and $y(t)$ representing the $x$ and $y$ coordinates of the particle's position, respectively. The first two terms are mean, $\mu_{\delta_{\mathbf{r}}} = \langle \delta\mathbf{r}(t) \rangle$, and variance, $\sigma_{\delta\mathbf{r}}^2 = \langle (\delta\mathbf{r}(t) - \mu_{\delta_r})^2 \rangle$, (first two moments) of the displacement distribution, $\Psi(\delta\mathbf{r}(t))$, respectively, where $\langle \cdot \rangle$ denotes the mean value over frames of a single trajectory. Trajectories of nanoparticles moving in LPTEM are often characterized by non-Gaussian displacement distributions[6,8]. This is notable during events when particles escape potential wells after being locally trapped with minimal displacements, resulting in large displacements in either negative or positive directions, which makes the distribution heavy-tailed and skewed. To accommodate these events, we incorporated skewness, $\frac{\langle (\delta\mathbf{r}(t) - \mu_{\delta r})^3 \rangle}{\langle \sigma_{\delta r} \rangle^3}$, and

kurtosis, $\frac{\langle(\delta\mathbf{r}(t)-\mu_{\delta\mathbf{r}})^4\rangle}{\langle\sigma^2_{\delta\mathbf{r}}\rangle^2}$,-related to the third and fourth moments of the distribution, respectively-into the loss function, where $\sigma_{\delta\mathbf{r}}$ is the standard deviation of $\Psi(\delta\mathbf{r}(t))$. Skewness addresses the asymmetry of the distribution, while kurtosis provides insight into the heaviness of the tail of the distribution. The four moments together can describe displacement distributions of trajectories with Gaussian or non-Gaussian statistics.

Two other key statistical measures that we incorporated into the loss function are the velocity autocorrelation function and positional autocorrelation function of the trajectories, $C_v(\tau)$, and $C_r(\tau)$, respectively. Here, $C_v(\tau) = \frac{\langle\mathbf{v}(t)\cdot\mathbf{v}(t+\tau)\rangle}{\langle\mathbf{v}^2(t)\rangle}$, where $\mathbf{v}(t)$ and $\mathbf{v}(t+\tau)$ are the velocities of the particle at time $t$ and $t+\tau$, respectively, and $\langle\,\cdot\,\rangle$ denotes the mean value calculated over time delays, $\tau$, for a single trajectory. $C_r(\tau)$ is defined as $C_r(\tau) = \frac{\langle\mathbf{r}(t)\cdot\mathbf{r}(t+\tau)\rangle}{\langle\mathbf{r}^2(t)\rangle}$, where $\mathbf{r}(t)$ and $\mathbf{r}(t+\tau)$ represent the position vectors at time $t$ and $t+\tau$, respectively. The $C_\mathbf{v}(\tau)$ captures dynamics not evident from the displacement distribution alone. The presence of the velocity autocorrelation function term in the loss function ensures that the model captures the anticorrelated motion of nanoparticles interacting with viscoelastic environments, such as those in crowded media, a well-known mechanism leading to anomalous motion[50]. Anticorrelation in particle trajectories is characterized by a negative value of the autocorrelation function at short time delays, $\tau$. We also calculate the ensemble velocity autocorrelation for a batch of trajectories to quantify the degree to which particle displacements are temporally correlated. The positional autocorrelation function term, $C_r(\tau)$, quantifies the spatial correlation between particle positions at different time delays. The inclusion of this term was motivated by the distinct behaviors observed in our experimental trajectories, where particles often transition abruptly between positions and remain localized in the new positions for extended periods. These dynamics introduce long-term spatial correlations that are not fully captured by displacement-based metrics. In addition to the autocorrelations, it is necessary to incorporate a term that accounts for the correlations between the $x$ and $y$ components of the 2-D trajectories. This term measures the correlation coefficient between displacements in the $x$ and $y$ directions, ensuring that the model captures any anisotropy or coupling between orthogonal motion components (see Supplementary Information for more details).

To further enhance LEONARDO's ability to capture the dynamics of experimental trajectories from LPTEM, we investigated the addition of other statistical terms inspired by the extensive set of statistical features discussed in the literature previously[51], which characterizes anomalous diffusion processes. Among these, which include the four moments of the displacements distribution, median of displacements, moments of the discrete Fourier transform, power spectral densities, and wavelet coefficients, we selected the median of displacements as an additional term in the loss function due to its positive effect on model performance based on the metrics discussed in Section 2.2. Detailed derivations of each term in the loss function are provided in the Supplementary Information, along with the training and validation losses of all terms at each epoch (See Fig. S1).

## LEONARDO model performance

A key goal of this study is to evaluate how well LEONARDO generates trajectories that resemble the experimental trajectories from LPTEM. For this purpose, we designed two performance metrics to quantitatively assess the similarity between LEONARDO-generated trajectories and experimental trajectories. These metrics not only provide a way to assess LEONARDO's performance but also assign a quantitative value to the similarity between other stochastic processes.

**Fréchet Distance between learned feature vectors.** The first metric we designed is based on the Fréchet Inception Distance (FID), a

measure widely recognized for its ability to compare distributions of multiple data points in a high-dimensional feature space originally introduced by Heusel et al.[52] to evaluate the quality of a given generative model in generating new image data. FID calculates the Wasserstein-2 distance between two multivariate Gaussian distributions, one representing real data and the other representing model-generated data. By comparing distributions of multiple samples, FID provides a global perspective on the statistical similarities between datasets. The Fréchet distance measures the mean and covariance differences between two distributions (see the Methods section for details), with lower scores indicating greater similarity between the two distributions. In the context of images, the distributions used in the FID method are the features extracted from the second-last layer of the Inception-v3 classifier[53], which is a 2048-dimensional vector representing high-level features of images, such as shapes, textures, and relationships of objects in images, rather than low-level pixel-level details. For our case of spatiotemporal trajectories, we utilized MoNet, a deep learning-based classifier developed specifically to analyze and characterize anomalous diffusion processes[6]. MoNet uses a dilated convolutional neural network to extract high-level features that differentiate between diffusion classes, making it a suitable substitute for Inception-v3 in this domain. The original MoNet was designed to classify three diffusion classes[6]. Herein, we adapted it into MoNet2.0 (Fig. 2c), which now classifies seven different diffusion classes and ensures capturing a broader range of stochastic processes, analogous to Inception-v3.

To train MoNet2.0, we used a diverse dataset comprising LPTEM trajectories, Brownian Motion (BM), and five classes of anomalous diffusion processes simulated using the models from the Anomalous Diffusion (AnDi) challenge[33] without any additional noise (see Methods Section for more details): FBM, CTRW, Annealed Transient Time Motion (ATTM), Scaled Brownian Motion (SBM), and Lévy Walk (LW) (Fig. 2a). We trained MoNet2.0 on 8000 trajectories per class, achieving a high accuracy across most classes as shown in Fig. 2d, with a high F1 score of 0.88 (a metric that balances precision and recall-see Methods section for the complete definition).

Using the trained MoNet2.0 model, we input 3202 LPTEM and 3202 LEONARDO-generated trajectories and extracted the second-last layer output of MoNet2.0 for each case-a 128-dimensional feature vector, analogous to the 2048-dimensional feature vector of Inception-v3 (Fig. 2b, c, e). We then calculated the Fréchet distance (FD) between these vectors. The resulting score of 7.88 demonstrates a high degree of fidelity between the LEONARDO-generated and experimental LPTEM trajectories. For context, the scores between different diffusion classes ranged from ~14.48 to ~57.58 (Fig. 2e). The scores computed between two independent batches of the same diffusion classes (e.g., FBM vs. FBM, CTRW vs. CTRW, etc.) range from 0.19 to 1.74 (see Fig. S4), providing a baseline for interpreting the FD values. These values highlight that a distribution of LEONARDO-generated trajectories is statistically closer to a distribution of LPTEM trajectories than to other classes of diffusion, and that the similarity between LPTEM and LEONARDO-generated trajectories is greater than the similarity between any other anomalous diffusion classes, further validating LEONARDO's generative performance.

Although we developed this FD metric specifically for evaluating the performance of LEONARDO, this metric provides a robust and versatile framework for quantifying the similarity between time-series processes, making it broadly applicable for the community to evaluate the performance of generative AI models for time-series data.

We also visualized the 128-dimensional feature vectors using the Universal Manifold Approximation and Projection (UMAP) in Fig. 2f. The UMAP embedding shows that the trajectories generated by LEONARDO and the real LPTEM experimental data occupy overlapping regions in the feature space, reinforcing the statistical similarity between the real experimental and the LEONARDO-generated data.

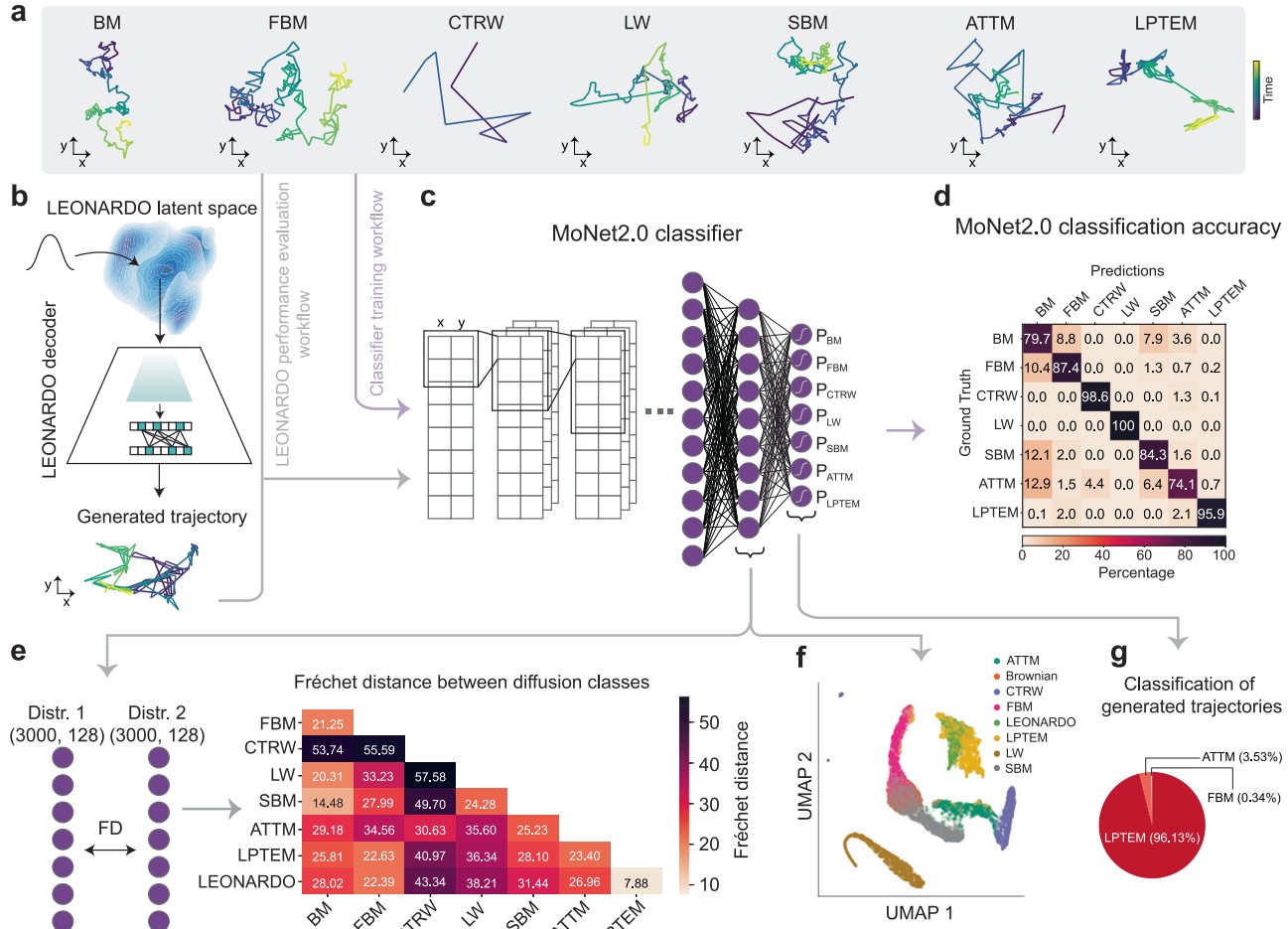

**Fig. 2 | Evaluation of the model performance of LEONARDO. a** Representative trajectories of different diffusion classes, including Brownian Motion (BM), five anomalous diffusion classes (FBM, CTRW, LW, SBM, and ATTM), and experimental trajectories from LPTEM. **b** Schematic illustrating the generation of trajectories by LEONARDO. Trajectories are sampled from a Gaussian distribution in LEONARDO's latent space and decoded to produce synthetic trajectories. **c** Schematic of the MoNet2.0 classifier architecture, trained using the trajectories from (**a**). To evaluate the performance of LEONARDO, trajectories from both (**a**) and (**b**) are input into the trained MoNet2.0 classifier. **d** Confusion matrix summarizing the classification performance of the MoNet2.0 model on benchmark diffusion classes, demonstrating high accuracy across most classes. **e** Schematic and results of the Fréchet Distance (FD) calculation. The second-last layer output of MoNet2.0 is used to compute the FD scores between pairs of diffusion class distributions. The lower triangular matrix shows that the FD score between LPTEM and LEONARDO-generated trajectories is significantly lower than the scores between other diffusion classes. **f** UMAP of the second-last layer of MoNet2.0 showing significant overlap between LPTEM and LEONARDO-generated trajectories. **g** Classification results of LEONARDO-generated trajectories by MoNet2.0 show that over 95% of LEONARDO-generated trajectories are classified as LPTEM, with smaller fractions classified as ATTM (3.53%) and FBM (0.34%).

**Classification of generated trajectories.** While the FD metric evaluates the similarity between the distributions of multiple trajectories, we also used the predicted probabilities of different classes of diffusion as outputted by the MoNet2.0 classifier as a second performance metric. By examining how each trajectory is classified by MoNet2.0, we gain insight into whether the generated trajectories capture the distinguishing features of the real experimental LPTEM trajectories. To evaluate this, we passed LEONARDO-generated trajectories through the trained MoNet2.0 classifier and recorded their predicted diffusion classes. More than 96% of the LEONARDO-generated trajectories were classified as LPTEM (Fig. 2g). This result highlights the strong resemblance between the generated and experimental trajectories, indicating that the LEONARDO-generated trajectories are more similar to LPTEM than any other class of diffusion, while a small percentage (less than 4%) of trajectories exhibit more resemblance to ATTM and FBM than LPTEM.

We note that while both the classifier and the FD are derived from the same MoNet2.0 model, they serve different purposes and are not expected to align exactly. The classifier assigns labels based on the most discriminative features in the learned feature space, and therefore highlights whether two trajectories can be separated by a sharp decision boundary. In contrast, FD measures the global similarity between two distributions across the entire 128-dimensional feature space. As a result, two classes may be close in terms of overall feature distributions (i.e., low FD), yet still be confidently separable by the classifier. This is seen, for example, in the case of Lévy Walk and Brownian Motion, which have a relatively low FD but are never confused in the classification task (Fig. 2d). Taken together, the FD and classification metrics offer complementary perspectives, and both indicate that LEONARDO-generated trajectories closely resemble the experimental LPTEM trajectories.

**Quantitative comparison of statistical properties.** In addition to FD and classification-based evaluation metrics, we further evaluated the statistical fidelity of LEONARDO's outputs by comparing both reconstructed and generated trajectories to experimental LPTEM trajectories. Specifically, we computed the ensemble-averaged values of each statistical metric used in the physics-informed loss function over 3202 trajectories (validation dataset size) and calculated the squared differences between these averages. For LEONARDO-generated

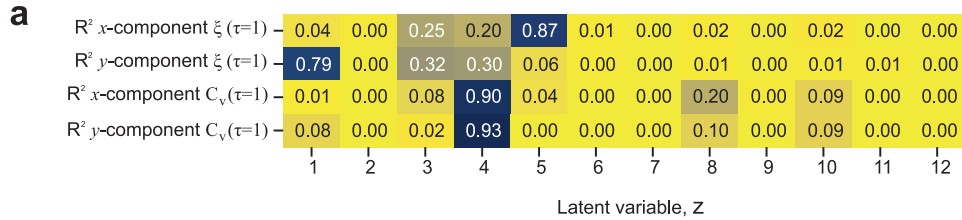

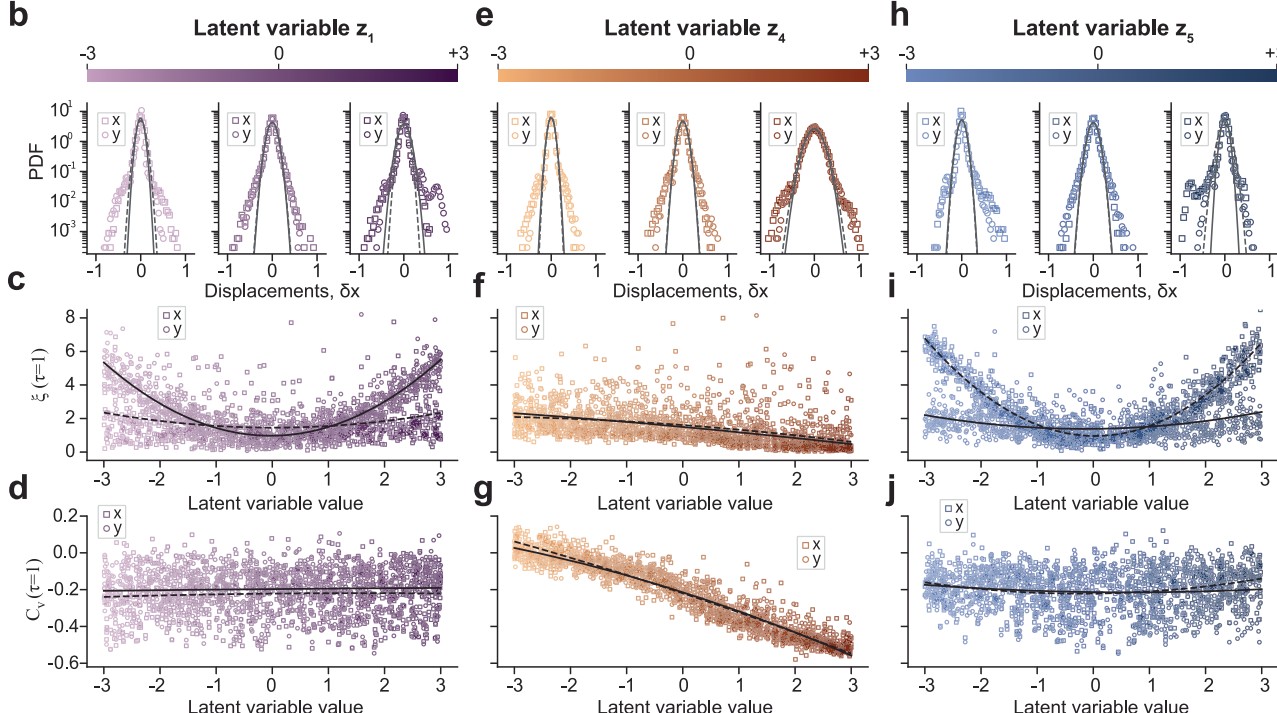

**Fig. 3 | Analysis of the physics learned by LEONARDO's latent variables. a** Matrix showing the coefficient of determination ($R^2$) for a second-degree polynomial fit between each latent variable $z_i$ ($i = 1, \ldots, 12$) and key statistical features of the trajectories: non-Gaussianity ($\xi(\tau = 1)$) of the $x$ and $y$ components (rows 1 and 2, respectively) and velocity autocorrelation ($C_v(\tau = 1)$) of the $x$ and $y$ components (rows 3 and 4, respectively). **b** Probability density functions (PDF) of displacement distributions of trajectories generated by LEONARDO by modulating the sampled value of latent variable $z_1$ from $-3$ to $+3$, corresponding to $\pm 3$ standard deviations away from the mean of the latent variable distribution. The PDFs of the $x$ and $y$ components are overlaid with Gaussian fits (dashed gray line for $x$ and solid gray line for $y$). **c** Scatter plot of non-Gaussianity ($\xi(\tau = 1)$) versus latent variable $z_1$. The black dashed and solid lines represent second-degree polynomial fits to the $x$ and $y$ components, respectively. **d** Scatter plot of velocity autocorrelation ($C_v(\tau = 1)$) versus latent variable $z_1$, with second-degree polynomial fits as in (**c**). **e–g** Same analysis as (**b–d**), but for latent variable $z_4$. **h–j** Same analysis as (**b–d**), but for latent variable $z_5$.

trajectories sampled randomly from the latent space, we measured how their average statistical properties differ from those of experimental LPTEM trajectories. As a reference point, we reported the same difference between the LEONARDO-generated trajectories and six different diffusion classes: Brownian motion, FBM, CTRW, Lévy Walk, ATTM, and SBM. We then evaluated LEONARDO-reconstructed trajectories, obtained by inputting experimental LPTEM trajectories into the trained model, and compared them statistically to the original LPTEM inputs. As shown in Table S1, LEONARDO-generated trajectories exhibit lower total weighted and unweighted squared errors against LPTEM compared to any reference diffusion class. The comparison between LEONARDO-reconstructed and original LPTEM input data demonstrates that LEONARDO can accurately recover the statistical properties of experimental trajectories, yielding significantly lower weighted and unweighted squared differences compared to reference comparisons between LEONARDO-generated trajectories and experimental LPTEM data, or simulated diffusion classes.

### Learning the underlying physics of trajectories

To investigate how LEONARDO's latent space encodes key statistical properties of the experimental data related to the interaction energy

landscape of nanoparticles with the LPTEM environment, we analyzed two specific metrics: the non-Gaussianity of the displacement distribution at $\tau = 1$ and the velocity autocorrelation at $\tau = 1$ (Fig. 3). The non-Gaussianity parameter, $\xi(\tau)$, of a given displacement distribution[54] is defined in terms of the fourth and second time-averaged moments for an ensemble of trajectories over a time delay window of size $\tau$ as $\xi(\tau) = \frac{1}{3}\frac{\langle \delta \mathbf{r}^4(\tau)\rangle}{\langle \delta \mathbf{r}^2(\tau)\rangle^2} - 1$[55], where $\langle \cdot \rangle$ denotes the average over an ensemble of trajectories. Non-Gaussianity in the displacement distribution can arise in systems where particles interact with heterogeneous potential wells, leading to localized trapping and intermittent escape events. A negative velocity autocorrelation, $C_v(\tau)$, at short time delays is often indicative of the caging effect, where viscoelastic forces in the medium impose directional constraints on particle motion over short time-scales. These statistical properties, although not direct evidence, serve as potential indicators of the complexity of the environment, offering a window into the heterogeneous and viscoelastic nature of the surroundings. Figure 3a shows the coefficient of determination ($R^2$) for the relationship between each latent variable and the non-Gaussianity and velocity autocorrelation of the $x$ and $y$ components of the trajectories at $\tau = 1$ (see Methods section for details about the calculation of $R^2$). We

generated trajectories by modulating each latent variable between $-3$ and $+3$ (corresponding to $\pm 3$ standard deviations from the mean), while all other latent variables were kept constant at their initial values sampled from a Gaussian distribution with a mean of zero and standard deviation of one. The statistical properties were then calculated for all trajectories. To capture potential non-linear relationships between latent variables and statistical properties, a second-degree polynomial was then fit to the data to calculate the $R^2$ values that quantify these relationships. The results in Fig. 3a suggest that latent variables $z_1$, $z_4$, and $z_5$ have strong relationships with the desired statistical properties, as quantified by the $R^2$ values, which are further explored in Fig. 3c, d, f, g, i, and j.

For latent variable $z_1$, as its sampled value is modulated from $-3$ to $+3$, the probability density function (PDF) of displacement distributions for the $y$-components of the generated trajectories (1000 trajectories per case) transitions from negatively skewed to positively skewed (Fig. 3b). The presence of heavy tails on the skewed side of the distributions contributes to an increase in non-Gaussianity. This change is reflected in $\xi(\tau)$ going from 6, indicating a non-Gaussian distribution, to near 0, representing a more Gaussian distribution, and back to 6 by increasing the value of $z_1$ (Fig. 3c). A similar trend is observed for the latent variable $z_5$ in the $x$-component, as shown in Fig. 3h, i. The high non-Gaussianity of displacement distributions is a common characteristic of trajectories where particles traverse a heterogeneous interaction energy landscape, experiencing prolonged entrapment within potential wells and infrequent escape events[50].

In Fig. 3d, j, corresponding to $z_1$ and $z_5$, there is little to no apparent relationship between the values of the latent variables and the velocity autocorrelation at $\tau = 1$. This indicates that $z_1$ and $z_5$ do not capture temporal correlations. In contrast, Fig. 3g, corresponding to $z_4$, exhibits a clear linear trend in both $x$ and $y$ components, highlighting a strong correlation between $z_4$ and the velocity autocorrelation at $\tau = 1$. However, Fig. 3e, f show that $z_4$ has a weak relationship with non-Gaussianity, suggesting that $z_4$ predominantly captures temporal correlations of trajectories.

Thus, $z_1$ and $z_5$ primarily learn the non-Gaussianity at $\tau = 1$, which is suggestive of the heterogeneity of the energy landscape that the particles traverse. Similarly, $z_4$ captures the velocity autocorrelation at $\tau = 1$, which is indicative of the viscoelasticity of the environment surrounding the particles. These results demonstrate how LEONARDO's latent variables capture statistical properties that relate to the physical processes governing particle motion in LPTEM. This capability allows LEONARDO to probe how experimental factors, such as particle size and electron beam dose rate, influence the interaction energy landscape in the liquid cell, as discussed in the next section. While we focused on three latent variables due to their relevance to the heterogeneity of the energy landscape and the viscoelasticity of the LPTEM environment, other latent variables also encode statistical properties important for capturing the full diversity of the physical processes in LPTEM. The complete table of the variances for all latent variables, computed from the encoder outputs on the test set, is provided in Table S2 of the Supplementary Information.

## The effect of particle size and electron beam dose rate of the microscope on the interaction energy landscape in LPTEM

Two key physical features that LEONARDO learned about the LPTEM trajectories through the latent space encoding of the trajectories are closely related to the interaction energy landscape that the nanoparticles traverse as they move near the surface of the $SiN_x$ membrane window of the liquid cell. As shown in Fig. 3, latent variables $z_1$, $z_4$, and $z_5$ are correlated with non-Gaussianity, related to the heterogeneity of this landscape, and anticorrelation, related to the viscoelasticity of the environment. Here, we leveraged LEONARDO's capabilities to explore the nature of the interaction energy landscape within the liquid cell environment. Specifically, we investigated how this landscape evolves

with the electron beam dose rate of the microscope and the nanoparticle size on new, unseen experimental data. To achieve this, we encoded trajectories from different experimental conditions and analyzed the distributions of the means ($\mu$) of the latent variables $z_1$, $z_4$, and $z_5$, as $\mu_1$, $\mu_4$, and $\mu_5$, respectively. Figure 4 shows the analysis for the trajectories collected at low and high dose rates (20 and 35 e$^-$/Å$^2 \cdot$ s) and for two different nanoparticle sizes (40 nm and 60 nm in length), where the resulting distributions of latent $\mu$ were plotted to examine how experimental factors influence the learned representations. The experimental data consisted of eight 4000-frame trajectories for three separate experimental conditions: (1) Dose 20, Size 60 (Fig. 4a–b, e–f, and i–k), (2) Dose 35, Size 60 (Fig. 4a–n), and (3) Dose 35, Size 40 (Fig. 4c–d, g–h, l–n). Each experimental trajectory was segmented into short (200-frame) pieces and encoded into LEONARDO to obtain the mean, $\mu$, of the latent variables for each piece. The distributions of these $\mu$ values were plotted for each experimental condition (Fig. 4a–d). Furthermore, we plotted the corresponding distributions of non-Gaussianity and velocity autocorrelation at $\tau = 1$ for the same datasets, which shows the relationship between the latent variables and the statistical properties for these experimental data (Fig. 4e–h). To provide a global view of how the different experimental conditions are encoded in the latent space, we also show the UMAP embeddings of the mean values of all twelve latent variables in Fig. 4i–n.

Figure 4a shows the distribution of the average of $\mu_1$ and $\mu_5$, $\langle \mu_1, \mu_5 \rangle$, for trajectories collected at dose rates of 20 and 35 e$^-$/Å$^2 \cdot$ s for a fixed particle size of 60 nm. Since both $\mu_1$ and $\mu_5$ encode non-Gaussianity in the $y$ and $x$ components, respectively (Fig. 3c, i shows this connection), we take their average, $\langle \mu_1, \mu_5 \rangle$, to capture the overall non-Gaussianity of the trajectories. At the lower dose rate of 20 e$^-$/Å$^2 \cdot$ s, the distribution is narrower with mainly lower values of $\langle \mu_1, \mu_5 \rangle$. In contrast, at a higher dose rate of 35 e$^-$/Å$^2 \cdot$ s, the distribution broadens, encompassing both higher and lower values of $\langle \mu_1, \mu_5 \rangle$. This observation is supported by the corresponding non-Gaussianity, $\xi(\tau = 1)$, distributions in Fig. 4e, where we show the computed non-Gaussianity averaged over the $x$ and $y$ components of trajectories. At the lower dose rate of 20 e$^-$/Å$^2 \cdot$ s, this distribution is narrower with lower values, while at the higher dose rate of 35 e$^-$/Å$^2 \cdot$ s, it becomes broader with both low and high values. This alignment reinforces the connection between $\langle \mu_1, \mu_5 \rangle$ and non-Gaussianity. Figure 4c compares the distribution of $\langle \mu_1, \mu_5 \rangle$ for particle sizes of 40 nm and 60 nm, both collected at a dose rate of 35 e$^-$/Å$^2 \cdot$ s. For the smaller particle size, the distribution is significantly narrower compared to the larger particle size, with most values concentrated around smaller magnitudes. However, interestingly, there are outliers reaching values less than $-2$ for the 40 nm-long particles. This finding is further corroborated by the corresponding non-Gaussianity distributions in Fig. 4g, where most values remain low for the smaller particle size, but rare outliers extend to higher non-Gaussianity. This consistency further reinforces the relationship between $\langle \mu_1, \mu_5 \rangle$ and non-Gaussianity. The observation of high values of non-Gaussianity at the higher dose rate of 35 e$^-$/Å$^2 \cdot s$ for both particle sizes suggests that dose rate plays a critical role in driving these extreme values, potentially contributing more to a heterogeneous energy landscape than the particle size.

We next investigated the latent variable $z_4$, which is strongly correlated with anticorrelation of both $x$ and $y$ components of trajectories, as shown in Fig. 3g. Figure 4b compares the distributions of $\mu_4$ for the dose rates of 20 and 35 e$^-$/Å$^2 \cdot$ s and a fixed particle size of 60 nm. At the lower dose rate, the distribution is narrower and shifted toward higher values, indicating stronger anticorrelation at lower dose rates. In contrast, at the higher dose rate of 35, the distribution broadens, spanning a wider range of values. This shift in the peak position suggests the presence of anticorrelation indicative of higher viscoelasticity at lower electron beam dose rates, consistent with previous observations reported in the literature[6,9]. The trend of anticorrelation with the electron beam dose rate is further

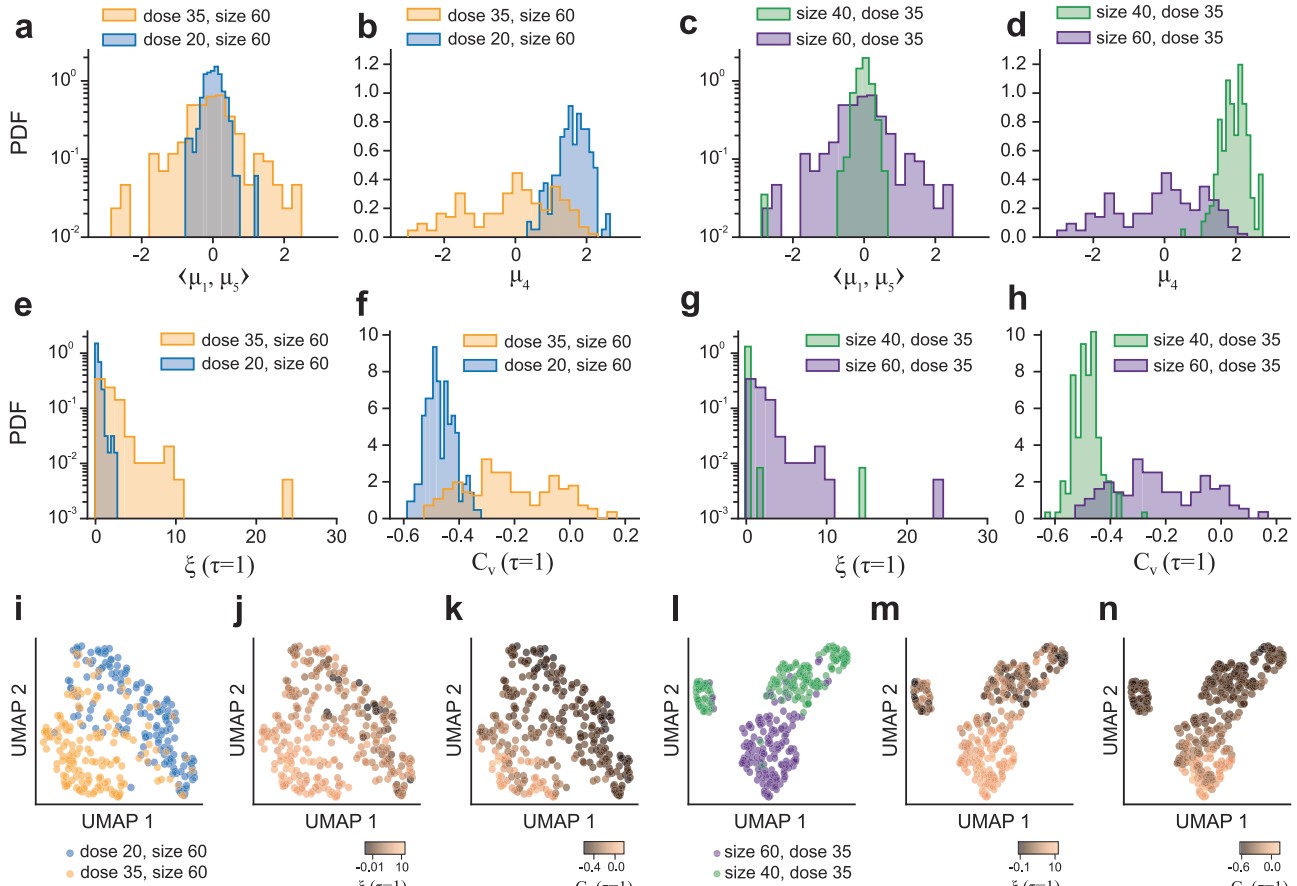

**Fig. 4 | Characterization of the interaction energy landscape in LPTEM using LEONARDO. a** Probability densities of the values of $\langle\mu_1, \mu_5\rangle$ for experimental trajectories collected at electron beam dose rates of 20 and 35 $e^-/\text{Å}^2 \cdot$ s for particle size of 60 nm. **b** Probability densities of the values of $\mu_4$ for the same experimental trajectories as in (**a**). **c** Probability densities of the values of $\langle\mu_1, \mu_5\rangle$ for experimental trajectories collected at particle sizes of 40 nm and 60 nm at an electron beam dose rate of 35 $e^-/\text{Å}^2 \cdot$ s. **d** Probability densities of the values of $\mu_4$ for the same experimental trajectories as in (**c**). **e** Probability densities of non-Gaussianity (averaged over the $x$ and $y$ components) at $\tau = 1$ for the same experimental trajectories as in (**a**). **f** Probability densities of the values of velocity autocorrelation (averaged over the $x$ and $y$ components of trajectories) at $\tau = 1$ for the same experimental trajectories as in (**b**). **g** Probability densities of non-Gaussianity at $\tau = 1$ for the same experimental trajectories as in (**c**). **h** Probability densities of the values of velocity autocorrelation at $\tau = 1$ for the same experimental trajectories as in (**d**).

**i** UMAP embeddings of all twelve latent variables for the same experimental trajectories as in (**a**) and (**e**), color-coded by dose rate. **j** UMAP embeddings of all twelve latent variables for the same experimental trajectories as in (**a**), (**e**), and (**i**), color-coded by non-Gaussianity ($\tau = 1$) on a symmetric logarithmic scale (Sym-LogNorm) **k** UMAP embeddings of all twelve latent variables for the same experimental trajectories as in (**a**), (**e**), and (**i**), color-coded by velocity autocorrelation ($\tau = 1$). **l** UMAP embeddings of all twelve latent variables for the same experimental trajectories as in (**c**) and (**g**), color-coded by particle size. **m** UMAP embeddings of all twelve latent variables for the same experimental trajectories as in (**c**), (**g**), and (**l**), color-coded by non-Gaussianity ($\tau = 1$) on a symmetric logarithmic scale (SymLogNorm) **n** UMAP embeddings of all twelve latent variables for the same experimental trajectories as in (**c**), (**g**), and (**l**), color-coded by velocity autocorrelation ($\tau = 1$).

corroborated by the distribution of the velocity autocorrelation (averaged over $x$ and $y$ components of trajectories) in Fig. 4f, which exhibits a similar shift. Figure 4d examines the effect of particle size on $z_4$ at a fixed dose rate of 35 $e^-/\text{Å}^2 \cdot$ s. For the smaller particles, the distribution is much narrower and centered around a $z_4$ value of 2, suggesting a much higher anticorrelation. This finding is supported by the distribution of velocity autocorrelation in Fig. 4h.

To further visualize how these dose-dependent and size-dependent effects manifest in the latent space, we examined the UMAP embeddings of all twelve latent variables. Figure 4i shows the UMAP representation color-coded by the electron beam dose rate, demonstrating a clear separation between the two experimental conditions. Figure 4j, k shows that this separation aligns with trends in non-Gaussianity and velocity autocorrelation, reinforcing the connection between these statistical properties and the learned latent variables. Similarly, Fig. 4l–n present the UMAP embeddings color-coded by particle size, non-Gaussianity, and velocity autocorrelation, respectively. The UMAP embeddings reveal a clear distinction between

the trajectories of 40 nm and 60 nm particles and their statistical properties, demonstrating that LEONARDO encodes differences in particle size through its learned statistical features. To further illustrate the relationships between the latent variables, pairwise scatter plots of $\mu_1$, $\mu_4$, and $\mu_5$ under the two dose rates and particle sizes are provided in Supplementary Fig. S6.

The possible viscoelasticity effect observed at lower electron beam dose rates may originate from the higher extent of elastic interactions with the $\text{SiN}_x$ membrane of the liquid cell chamber. These interactions arise due to multiple forces on nanometer length scales, including van der Waals forces, electrostatic interactions, and adhesion to the membrane, which create interaction potential wells with binding stiffness $k$ that locally trap or cage nanoparticles, opposing their motion and restoring them to their current positions[56]. A similar but stronger trend is observed for particle size, where smaller particles exhibit narrower distributions of $\mu_4$ and higher anticorrelation, suggesting a more localized trapping effect. In contrast, larger particles display a broader range of $\mu_4$ values, indicating weaker caging effects and a wider distribution of

elastic interactions. These observations point to a plausible relationship between particle size, dose rate, and viscoelasticity: the extent of caging and trapping is modulated both by the size of the particle and by the dose rate of the electron beam, with smaller particles and lower dose rates experiencing more pronounced viscoelastic effects.

Overall, these results demonstrate how LEONARDO's latent variables $z_1$, $z_4$, and $z_5$ capture the effects of experimental parameters, such as electron beam dose rate and particle size, on the underlying statistical properties of trajectories, which are potentially related to the interaction energy landscape of the liquid cell environment. High electron beam dose rates are associated with greater non-Gaussianity and, therefore, increased heterogeneity, regardless of particle size. In contrast, smaller particles and lower electron beam dose rates exhibit stronger anticorrelation, indicative of enhanced viscoelastic effects within the liquid cell environment.

In summary, we reported developing a new data-driven model for learning the stochastic motion of nanoparticles in LPTEM experiments using a transformer-based VAE model named LEONARDO. The novelty of LEONARDO is a customized loss function informed by the physics of nanoparticle trajectories and an attention-based transformer that enables learning time dependencies of particle trajectories. LEONARDO is trained on a large dataset of trajectories from LPTEM experiments collected in a range of electron beam dose rates and for short and long nanorods. The model captures the physics related to the interaction energy landscape surrounding the particle by probabilistically mapping the data into a low-dimensional space. We demonstrated that LEONARDO identifies characteristics of the interaction energy landscape in LPTEM by learning statistical properties associated with the energy landscape via three of the latent variables.

By applying LEONARDO to an unseen set of experimental LPTEM trajectories collected at varying electron beam dose rates and for nanoparticles of different sizes, we report that both dose rate and particle size jointly modulate key statistical features, such as non-Gaussianity and anticorrelation, which are indicative of heterogeneity and viscoelasticity in the liquid cell environment. These results highlight LEONARDO's ability to uncover statistical and physical insights into the stochastic motion of nanoparticles and its potential as a tool for analyzing complex particle-environment interactions. LEONARDO is capable of producing single particle trajectories, which can serve as a simulator model for LPTEM experiments, offering a cost-effective alternative to acquiring experimental data. We envision that such synthetic data can be used to train models for automating electron microscope-based workflows in the future.

## Methods

### Experimental method

**Liquid phase TEM silicon nitride chip preparation.** Microfabricated silicon nitride chips from Protochips Inc. with 550 µm × 50 µm and 550 µm × 20 µm window and with 50 nm and 150 nm spacers were used for encapsulating 0.5 microliters of the gold nanorod solutions. The chips were first cleaned by immersing them in an acetone bath, followed by dipping them into an ethanol bath, and subsequently dried by blowing nitrogen gas parallel to the surface. The surface of the chips was then made hydrophilic using easiGlow, the glow discharge cleaning system by PELCO, to ensure that the liquid solution spreads evenly across the surface of the chip. The settings used for easiGlow were 0.39 mBar pressure and a glow discharge time of 45 seconds. The chips were then assembled inside the Poseidon Select holder from Protochips before imaging on the TEM.

**TEM imaging.** The LPTEM experiments were performed on the FEI Tecnai F30 TEM of the Materials Characterization Facility of the Institute for Matter and Systems of the Georgia Institute of Technology operating at 200 kV. The gold nanorods were visualized using the Gatan OneView in situ camera, and videos of nanorod motion were recorded using the Digital Micrograph software. The electron beam dose rate used for each experiment was calibrated for camera magnifications of 19.5 kx, 25 kx, and 29.5 kx using a custom-built Digital Micrograph script. Various dose rates of 2 to 60 e$^-$/Å$^2 \cdot$ s were used to capture a range of different diffusion regimes of the gold nanorods. Videos were recorded at various camera exposures from 0.005 s to 0.1 s, corresponding to frame rates of 200 to 10 frames per second. Camera resolutions of 512 × 512, 1024 × 1024, and 2048 × 2048 pixels by pixels were used.

### Processing of in situ videos

The in situ videos (time series of frames) collected from the LPTEM experiments were processed to extract the positions of single particles. First, a custom MATLAB script was employed to extract a region of interest (ROI) from each video frame, which contained the complete length of a single particle trajectory. Subsequently, the series of ROIs from each video were processed using a thresholding-based algorithm to obtain the $x$ and $y$ coordinates of the centroid of each nanorod in each frame. Details on further processing of trajectories to generate the training dataset can be found in the Supplementary Information.

### LEONARDO model

LEONARDO (Learning Electron microscopy Of NAnopaRticle Diffusion via an attention netwOrk) is a transformer-based variational autoencoder with encoder and decoder blocks that use multi-headed attention mechanisms in the encoder and decoder to learn the time-dependent dynamics of the trajectories. In doing so, LEONARDO maps an input trajectory of 200-frames, $\mathbf{r} = (\mathbf{r}_1, \mathbf{r}_2, \cdots, \mathbf{r}_{T=200})$, where $\mathbf{r}_t = (x_t, y_t)$ represents the position vector of the nanoparticle at time $t$, with $x_t$ and $y_t$ denoting the $x$ and $y$ coordinates of the particle's position, respectively, to a sequence of continuous representation $\mathbf{z} = (z_1, \cdots, z_{12})$, which are the latent variables of the model. Given $\mathbf{z}$, the decoder generates an output trajectory $\hat{\mathbf{r}} = (\hat{\mathbf{r}}_1, \cdots, \hat{\mathbf{r}}_{200})$. The attention block in the encoder uses 8 heads and 2 layers with an embedding size of 128. The output of the attention block goes into a convolutional encoder. The convolutional encoder consists of two convolutional layers with kernel sizes of 7 and 2, respectively. Following the convolutional layers, a dense layer feeds the output into the latent space with a dimension of 12. The latent space is then upsampled using a dense layer and a transpose convolutional layer. The output from the convolutional layer is input into a multi-headed attention block in the decoder before being reshaped into the final output trajectory using a convolutional layer, which serves to preserve temporal correlations in the trajectories.

The training dataset consisted of 38,279 LPTEM trajectories, the validation dataset consisted of 3202 LPTEM trajectories, while the test dataset included 5934 LPTEM trajectories. The model was trained for 200 epochs, during which the validation set was used to tune hyperparameters, such as determining the optimal combination of loss function components based on model performance metrics. The final model performance metrics were reported using the test dataset.

Figure S1 of the Supplementary Information shows the plots of training and validation losses, and Figure S3 presents a detailed architecture of the LEONARDO model.

### Fréchet Distance

The Fréchet Distance, $d$, measures the mean and covariance differences between two distributions using the formula:

$$d^2 = \| \mu_r - \mu_g \|_2^2 + \mathrm{Tr}\,(\Sigma_r + \Sigma_g - 2(\Sigma_r \Sigma_g)^{1/2}), \tag{1}$$

where $\mu_r$ and $\Sigma_r$ are the mean and covariance of the real data features, and $\mu_g$ and $\Sigma_g$ are the mean and covariance of the generated data features, with lower scores indicating greater similarity between the two distributions.

**Theoretical stochastic diffusion processes and their simulation**
To train MoNet2.0, we generated a diverse dataset comprising six stochastic diffusion processes: Brownian Motion (BM), Fractional Brownian Motion (FBM), Continuous Time Random Walk (CTRW), Annealed Transient Time Motion (ATTM), Scaled Brownian Motion (SBM), and Lévy Walk (LW), which are subsequently normalized according to equation S1 in the Supplementary Information. Below, we provide a brief theoretical description for each stochastic process and describe the methods used to simulate each process. For full theoretical details of simulating the anomalous diffusion processes (FBM, CTRW, ATTM, SBM, and LW), readers are referred to the Anomalous Diffusion (AnDi) Challenge and the corresponding article[33].

**Brownian motion.** Brownian motion describes a purely random process, where the particles undergo thermal motion. The fundamental equation for Brownian Motion is:

$$\frac{\partial}{\partial t} P(x, t) = D \frac{\partial^2}{\partial x^2} P(x, t), \tag{2}$$

where $P(x, t)$ is the probability density function for the particle that describes the position of a particle in position $x$ at time $t$ and $D$ is the diffusion coefficient that is characteristic of the particle and its surrounding environment (particle geometry and surrounding temperature). Solving this equation with an initial condition of $x = 0$ at $t = 0$ (*i.e.*, $\Delta x_i = x_i$) with unbounded $x$ and $t$ results in:

$$P(x, t) = \frac{1}{\sqrt{4\pi D t}} \exp\left(-\frac{x^2}{4Dt}\right). \tag{3}$$

The first moment (mean) of this distribution is zero, $\langle x(t) \rangle = 0$, and the second moment of this distribution, *i.e.*, the variance $\langle x^2(t) \rangle$, that is also the ensemble-averaged mean squared displacement (e-MSD) of the trajectories coming from this distribution has the form:

$$\langle x^2(t) \rangle = \int_{-\infty}^{+\infty} x^2 \cdot \frac{1}{\sqrt{4\pi D t}} \exp\left(-\frac{x^2}{4Dt}\right) dx = 2Dt. \tag{4}$$

Trajectories were generated as discrete realizations of Brownian motion by summing Gaussian-distributed random displacements at each time step. For each trajectory, the displacements were sampled from a normal distribution with a zero mean and a variance proportional to the time step. The simulation iteratively updates the position as:

$$x_{i+1} = x_i + \mathcal{N}(0, 1), \quad y_{i+1} = y_i + \mathcal{N}(0, 1), \tag{5}$$

where $x_i$ and $y_i$ are the current positions, and the displacements $\mathcal{N}(0, 1)$ are sampled independently for $x$ and $y$ directions.

**Fractional Brownian motion.** FBM generalizes Brownian motion by introducing memory effects in the step increments, characterized by the Hurst exponent $H = \frac{\alpha}{2}$. The mean squared displacement (MSD) for particle position $x$ scales as:

$$\langle x^2(t) \rangle \sim t^\alpha, \quad 0 < \alpha < 2, \tag{6}$$

where $\alpha < 1$ corresponds to subdiffusion and $\alpha > 1$ to superdiffusion. FBM trajectories are generated using fractional Gaussian noise for both $x$ and $y$ directions. The process ensures that the increments $\Delta x$ and $\Delta y$ are correlated in time according to $H$:

$$x_{i+1} = x_i + \eta_{x,i}(H), \quad y_{i+1} = y_i + \eta_{y,i}(H), \tag{7}$$

where $\eta_{x,i}(H)$ and $\eta_{y,i}(H)$ are fractional Gaussian noise processes with memory effects determined by $H$. To train MoNet2.0, we sampled

values of $\alpha$ (or equivalently $2H$) uniformly in the subdiffusive range of [0.1, 1].

**Continuous time random walk.** CTRW introduces waiting times between particle steps, where the waiting times $\psi(\tau)$ follow a power-law distribution:

$$\psi(\tau) \sim \tau^{-(1+\alpha)}, \quad 0 < \alpha < 1. \tag{8}$$

This results in subdiffusive behavior, as particles remain trapped in localized regions for long periods before stepping. To simulate CTRW trajectories, waiting times are drawn from the power-law distribution. After each waiting time, the particle position is updated with independent Gaussian random steps in both the $x$- and $y$-directions. The cumulative positions are then regularized to equally spaced time intervals. Specifically, the particle positions at time frame $i + 1$ are iteratively updated as:

$$x_{i+1} = x_i + \eta_{x,i}, \quad y_{i+1} = y_i + \eta_{y,i}, \tag{9}$$

where $\eta_{x,i}$ and $\eta_{y,i}$ are independent Gaussian-distributed random steps applied after each waiting time. The resulting trajectories exhibit subdiffusion due to the broad distribution of waiting times. To train MoNet2.0, we sampled the values of $\alpha$ uniformly in the range of [0.1, 1].

**Lévy walk.** Lévy walks are stochastic processes that introduce long flights of constant velocity. The MSD scales as:

$$\langle x^2(t) \rangle \sim t^\alpha, \quad 1 < \alpha < 2. \tag{10}$$

Here, $\alpha$ controls the distribution of step durations, leading to superdiffusive behavior when $1 < \alpha < 2$. To simulate Lévy walk trajectories, the step durations $\tau_i$ are drawn from a power-law distribution:

$$\phi(\tau) \sim \tau^{-(4-\alpha)}, \quad 1 < \alpha \le 2, \tag{11}$$

with the exponent defined by $\sigma = 3 - \alpha$ (using a random $\sigma$ when $\alpha = 2$). For each flight, a constant velocity, $v$, is sampled and a single flight direction, $\theta$, is drawn uniformly from $\theta = [0, 2\pi]$; the flight is discretized into an integer number of time steps, during which the particle positions are updated as

$$x_{i+1} = x_i + v \cos(\theta), \quad y_{i+1} = y_i + v \sin(\theta). \tag{12}$$

To train MoNet2.0, values of $\alpha$ were sampled uniformly from [1, 2].

**Annealed transient time model.** ATTM describes a process in which the diffusion coefficient $D$ fluctuates over discrete time intervals, leading to subdiffusive dynamics. For a given $D_i$ each time step is defined as

$$t_i = D_i^{-\gamma}, \tag{13}$$

and the particle position is updated as

$$x_{i+1} = x_i + \sqrt{2D_i t_i}\, \eta_{x,i}, \quad y_{i+1} = y_i + \sqrt{2D_i t_i}\, \eta_{y,i}, \tag{14}$$

where $\eta_{x,i}$ and $\eta_{y,i}$ are independent standard Gaussian random variables. To train MoNet2.0, values of $\alpha$ were sampled uniformly from [0.1, 1].

**Scaled Brownian motion.** SBM describes nonstationary diffusion in which the diffusion coefficient varies with time as

$$D(t) \sim t^{\alpha-1}, \tag{15}$$

so that the mean squared displacement scales as

$$\langle x^2(t) \rangle \sim t^\alpha. \tag{16}$$

To simulate SBM, the variance is given by $\sigma^2 t^\alpha$; thus, if the variance at time step $i$ is $\sigma^2 i^\alpha$, the position is updated as

$$x_{i+1} = x_i + \sqrt{\sigma^2 \left[ (i+1)^\alpha - i^\alpha \right]} \, \eta_{x,i}, \quad y_{i+1} = y_i + \sqrt{\sigma^2 \left[ (i+1)^\alpha - i^\alpha \right]} \, \eta_{y,i}, \tag{17}$$

where $\eta_{x,i}$ and $\eta_{y,i}$ are independent standard Gaussian variables. To train MoNet2.0, values of $\alpha$ were sampled uniformly from [0.1, 1].

Supplementary Fig. S5 shows the UMAP plot of the latent space of LEONARDO with AnDi simulated trajectories and experimental LPTEM trajectories encoded.

### Calculation of coefficient of determination ($R^2$)

To quantify the relationship between latent variables and statistical features, we used the coefficient of determination ($R^2$). This metric evaluates how well a model fits the data by comparing the variance explained by the model to the total variance of the data. Specifically, $R^2$ is calculated as:

$$R^2 = 1 - \frac{SS_{res}}{SS_{tot}} \tag{18}$$

where $SS_{res}$ is the sum of squared residuals:

$$SS_{res} = \sum_{i=1}^{n} \left( y_i - y_{fit,i} \right)^2 \tag{19}$$

and $SS_{tot}$ is the total sum of squares:

$$SS_{tot} = \sum_{i=1}^{n} \left( y_i - \langle y \rangle \right)^2 \tag{20}$$

Here, $y_i$ represents the observed data, $y_{fit,i}$ is the predicted value from the second-degree polynomial fit, and $\langle y \rangle$ is the mean of the observed data. $R^2$ values range from 0 to 1, with higher values indicating a stronger relationship between the variable of interest and the fit. This approach allows us to quantitatively assess the extent to which each latent variable captures the statistical properties of the trajectories in Section 2.3.

### Calculation of F1 score

The F1 score is a metric that combines precision and recall as follows:

$$F1 = 2 \cdot \frac{\text{Precision} \cdot \text{Recall}}{\text{Precision} + \text{Recall}} \tag{21}$$

where precision is given by:

$$\text{Precision} = \frac{\text{TP}}{\text{TP} + \text{FP}} \tag{22}$$

and recall is given by:

$$\text{Recall} = \frac{\text{TP}}{\text{TP} + \text{FN}} \tag{23}$$

Here, TP refers to the number of true positives, FP to false positives, and FN to false negatives. The final F1 score for MoNet2.0 was calculated by averaging the F1 scores across all diffusion classes as follows:

$$F1_{weighted} = \sum_{i=1}^{N} w_i \cdot F1_i \tag{24}$$

where $N$ is the total number of classes, $F1_i$ is the F1 score for class $i$, and $w_i$ is the weight for class $i$, given by:

$$w_i = \frac{n_i}{\sum_{j=1}^{N} n_j} \tag{25}$$

where $n_i$ is the number of trajectories in class $i$.

## Data availability

The gold nanoparticle trajectory data used in this study for training, validating, and testing the LEONARDO model, as well as the experimental datasets used in the electron beam dose and size study (Section 2.4), are deposited in the HuggingFace repository under accession code https://doi.org/10.57967/hf/5786. Source data are provided with this paper.

## Code availability

The LEONARDO and MoNet2.0 source code is available on GitHub at https://github.com/JamaliLab/LEONARDO and can be accessed and referenced via Zenodo at https://doi.org/10.5281/zenodo.15708218[57]. The trained LEONARDO model is available on HuggingFace at https://doi.org/10.57967/hf/5787.

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

## Acknowledgements

This research was supported by the NSF, Division of Chemical, Bioengineering, Environmental, and Transport Systems under award 2338466, the American Chemical Society Petroleum Research Fund under award 67239-DNI5, the Exponential Electronics seed grant of the Institute for Matter and Systems at Georgia Tech, and Georgia Institute of Technology start-up funds. N.G. acknowledges the support from the President's Undergraduate Research Award (PURA) at Georgia Tech. The authors thank Dr. Amirali Aghazadeh, Dr. Cory Hargus, and Daniel Saeedi for insightful discussions. The authors thank Dr. Nasrin Houshmand and the Laser Dynamics Laboratory and Dr. Brettmann and her group at Georgia Tech for allowing us to use their wet lab space for part of our nanoparticle synthesis. The authors also thank Dr. Yong Ding, Dr. Ben Miller, and Mr. Thomas Zhang for their help in implementing our dose rate control code with the Oneview camera at Georgia Tech. The authors acknowledge the support of the Material Characterization Facility and the Electron Microscopy Facility of the Institute for Matter and Systems at Georgia Tech, a member of the National Nanotechnology Coordinated Infrastructure (NNCI), which is supported by the National Science Foundation (ECCS-2025462).

## Author contributions

Z.S. and V.J. conceived the project and designed the experiments and the neural network architecture. Z.S. collected the data, developed the model, and wrote and implemented the code. N.G. implemented the image analysis code. P.A.N. synthesized and characterized nanoparticles. Z.S., N.G. and V.J. analyzed the data and generated the figures. Z.S., N.G., P.A.N. and V.J. wrote the manuscript. V.J. supervised the project.

## Competing interests

The authors declare no competing interests.
