## [Transparent Peer Review file · Nature Communications]

Learning the Diffusion of Nanoparticles in Liquid Phase TEM via Physics-informed Generative AI

Corresponding Author: Professor Vida Jamali

Version 0:

Reviewer comments:

Reviewer #1

(Remarks to the Author)

The authors present an unsupervised method to extract the physical features from the motion of nanoparticles in LPTM experiments. In particular, they use a VAE with a physics-inspired loss function that helps the model capture the properties of input trajectories. The work tackles an important topic: the autonomous extraction of new knowledge, in terms of physical features, directly from stochastic data, in an unsupervised way. The work is very well presented and the goals and methods to achieve them are clear. The associated code is useful for the experts and helped me check some details about the dataset and model use. It is however not suitable for non-experts, who may need further information or an “introductory” notebook.

I have few questions regarding the technical aspects of the work. To summarize what I expand in the comments below: it is not clear which latent features have been exactly learned by the VAE. I think that: 1) the VAE could have just learned the experimental properties being changed in the experiments (beam dose rate and nanorod size); 2) the simulated dataset added to the training may have biased the VAE towards the properties of FBM and CTRW. I hope that my questions below help clarify these points.

1. Major comments:

1.1) First, I would like to point out that a recent work (Fernandez-Fernandez et al. PRE Phys. Rev. E 110, L012102 (2024)) tackles the same problem with a VAE, although without a physics informed loss. However, in that case the “reconstruction” loss (i.e. L_1 of the authors loss) is changed to a probabilistic loss that compares the distribution of the decoder’s output to the distribution of the dataset. In this way, all terms considered by the authors for their physics informed loss are taken into account implicitly.

1.2) In that same work, the authors claim that an autoregressive decoder is needed in order to generate FBM trajectories, as only in this case the decoder’s output can show the proper correlations. While the authors show that the generated show some correlation via the autocorrelation plots, to which extent their model can actually generate trajectories with long range correlations as FBM ones? I suspect that while the transformer decoder may be close to do so to some extent, the linear layer at the end of the decoder may “kill” most of the correlations.

1.3) About the dataset, I fail to understand the purpose of augmenting it with simulated data. If the authors aim to extract the physical features of their experiment, wouldn’t it be enough to train with experimental trajectories? Following the author’s train of thought, if we are not sure about the true model describing the experiments, can not the simulated dataset wrongly bias the VAE latent? Have the authors trained a VAE purely with experimental data? What latent space would arise from there?

1.4) Related to the latter, I wonder what is exactly in the latent space of the VAE. The ML literature has shown us that these models are able to learn the factors of variations of a given dataset.

A) If focusing only on the experimental dataset, there are two factors of variation: the size of the nanorod and the beam dose

rates. I am not an expert on this experiment, but I understand that these properties will affect the diffusion coefficient and the anomalous exponent. I understand that the diffusion coefficient here is not a factor of variation due to the normalization procedure.

B) If considering that the simulated dataset has exactly the same properties as the experimental dataset, we have at least three factors of variation: 1) the value κ of Eq. S5; 2) the α of the CTRW; 3) the α of the FBM. Actually, are these two latter sample independently? I also think that in this case the generalized diffusion coefficient should enter into play as the normalization procedure may affect FBM and CTRW trajectories differently, as D_α depends on α in two different ways.

Both these cases make me wonder what exactly has the VAE learned? Is the VAE extracting the experimental features (size and dose) or the physical features (κ s, α s, D_α s). I think some ablations on the dataset (i.e. training only on experiments / simulated) may help discern this.

1.5) A common feature of VAEs is that the variance of most of the neurons will go to 1 (as given by the prior) while the neurons needed by the decoder to generate new data converge to something smaller. Have the authors checked this? In their case, only the neurons 2 and 12 should have small variances while the rest should go to 1.

1.6) Given the previous, all neurons but 2 and 12 are meaningless, as their variance is very big. This means that the effective latent manifold is of dimension 2. Hence, why is the TSNE in Figure 2 needed? Could not a similar plot be achieved by plotting z_2 vs z_{12} ?

1.7) While Fig.2b gives vital information to understand the latent space of the VAE, due to the number of points plotted and the colors for FBM and CTRW being so close to each other, it is hard to understand in detail what is happening. Maybe plotting the mean z will show the same in a clearer manner?

2. Minor comments:

2.1) The authors reference [39] as an unsupervised method but that method is supervised.

2.2) Given that the authors have a Transformer encoder, wouldn't it be possible to feed trajectories of all sizes, if not considering the current input Linear layer? This would be an upgrade to the model to avoid having to "cut" the trajectories (and I understand may fall out of the scope of this work)

2.3) If I understand correctly, the authors separated the two components of the trajectory to effectively create two trajectories. Wouldn't the model benefit from having as input the two coordinates as input (e.g. a tensor of size (2 x 200))?

2.4) There seems to be a typo in line 143: traction -> attraction?

To summarize, I think the VAE approach shows great potential for the study of anomalous diffusion. The authors make a great effort to apply it to experimental data. However, because they combine the training with simulated data, it is hard to dissociate exactly what has been learned in the latent space. I hope some of my concerns help clarify these points.

(Remarks on code availability)

The associated code is useful for the experts and helped me check some details about the dataset and model use. It is however not suitable for non-experts, who may need further information or an "introductory" notebook.

Reviewer #2

(Remarks to the Author)

The authors developed an interesting method to generate stochastic trajectories via a neural network architecture. The generated trajectories match some of the statistical properties of the ones the networks were trained on but are new, independent trajectories. The statistical properties chosen by the authors are the mean, variance, skewness, kurtosis of the displacements and the particle velocity autocorrelation. The neural network architecture involves an encoding part that compresses the input trajectory and encodes their salient features, followed by a decoding part that generates new trajectories possessing those encoded features. Interestingly, by inspecting the latent space where the features are encoded, it is possible to gain some physical insight into the input trajectories and to generate trajectories with similar characteristics. I find this approach interesting, and the attempt to increase the interpretability of neural networks for single particle trajectories is a much-needed and ambitious effort. However, I feel the manuscript provides mostly qualitative evidence, thereby falling short of fully reaching its goal of providing an interpretable machine-learning method. It also contains many bold claims that are only loosely motivated. I will provide my detailed criticism below.

MAIN COMMENTS

1. The authors do not provide convincing metrics about the performances of their method. It is then impossible for the reader to understand how "good" these models are at generating or analysing trajectories. How closely do the generated trajectories capture the experimental ones? (see my comment below as well). How reliable and accurate is the quantitative inference of the anomalous diffusion exponent contained in the latent variables? Many methods have been developed to estimate the anomalous exponent both using traditional [e.g., V. Tejedor, et al, Biophys. J. 98, 1364 (2010) K. Burnecki, et al, Sci. Rep. 5, 11306 (2015). Y. Meroz and I. M. Sokolov, Phys. Rep. 573, 1 (2015). N. Makarava, et al, Phys. Rev. E 84,

021109 (2011). A. Weron, et al Phys. Rev. E 99, 042149 (2019), J. H. Jeon and R. Metzler, J. Phys. A: Math. Theor. 43, 252001 (2010), Krapf, and M. M. Tamkun, Biophys. J. 111, 1235 (2016), G. Sikora, et al, Phys. Rev. E 96, 062404 (2017). A. Weron, et al, Sci. Rep. 7, 5404 (2017)] and AI tools. The authors should show how their method performs compared to them, for instance by using the AnDi dataset [Muñoz-Gil, G. et al. Nature Communications 12, 6253 (2021)], which provides an excellent benchmark.

2. The authors should justify why they chose the specific 5 statistical features mean, variance, skewness, kurtosis of the displacements and the particle velocity autocorrelation to assess the similarity of the trajectories. For example, the authors of A Gentili and G Volpe 2021 J. Phys. A: Math. Theor. 54 314003 used many more features (over 90) to properly assess the anomalous diffusive behaviour of single trajectories. How can the authors confidently claim that trajectories sharing their chosen 5 properties are effectively representing the same process, without resorting to more refined spectral properties of the signal and correlations? This analysis is necessary to confidently show that the trajectories are indeed accurate and realistic. Interesting works that could be helpful in this direction are, for instance, Krapf New J. Phys. 20 (2018) 023029, which analysed what can be learned from the power spectral density of a single Brownian trajectory and analysed anomalous diffusion in Krapf Phys. Rev. X 9, 011019

3. The authors state: "The short single-particle trajectories used to train Leonardo allow the analysis of any length of real experimental data by simply segmenting the longer experimental trajectories into shorter pieces and analyzing their time-evolving behavior". Very often, learning from short trajectories returns methods that are best suited for short trajectories and perform poorly on longer ones. The present method may suffer from the same shortcomings. The authors should comment on this issue.

4. Why did the authors choose to focus on trajectories containing 200 frames?

5. Why did the authors choose to split the data into 1D trajectories? The two spatial dimensions should be highly correlated and this splitting should be properly motivated.

6. It is not clear what the authors mean with "The loss function also incorporates an additional term that includes the mean value of the displacement calculated over a batch of trajectories in each training step, which ensures that large displacements are not taken at the same time points in every trajectory".

7. I find panels 2b and 2c unclear and rather difficult to read. Also, their discussion in the text (line 209) is not fully satisfactory.

8. Figure 3 reports very interesting results, contributing to the interpretability of the method. However, the discussion is overly stretched and not fully supported by the data. Figure 3b shows a flat profile, except for the values of 2 and 3. Similarly, Fig. 3e, shows little variation for positive values of the latent variable. It is then difficult to confidently claim that these latent variables adequately capture non-Gaussianity.

9. The statement at the end of section 2.2 "Thus, z_2 essentially captures the heterogeneity of the energy landscape of the environment that the particles interact with as they move, while z_{12} captures the local viscoelasticity of the environment surrounding the particles. This observation confirms that the latent variables of Leonardo learned specific physics about the nanoparticles and their interactions with their surrounding environment due to the implementation of the physics-informed loss function, enabling it to distinguish between processes of different mechanisms, including the complex motion of nanoparticles in LPTM." is quite speculative and should be stated as such. What the authors find are intriguing, suggestive indications and correlations between latent variables and some statistical features. These are definitely not hard evidence. The connection between the statistical features and the physical picture is very speculative. Therefore the authors' claims should be less confident and clearly stated to be, though perhaps plausible, speculations.

10. Figure 4, although interesting, is mostly a visual, non-quantitative one and its discussion is handwaving. Its scientific significance is, therefore limited.

11. In the beginning of section 2.4 the authors claim that two latent variables capture the non-Gaussianity and the negative autocorrelation of the trajectories. To convincingly show this the authors should provide more quantitative evidence than the one presented in Fig. 3. Scatter plots similar to the ones of Fig. 2b and correlation coefficients may serve this purpose well.

12. How would the plots in Fig. 5 look if non-Gaussianity and the magnitude of the negative peak of autocorrelation were plotted? This would help understand to what extent the correlation between the two latent variables and these statistical features, reported in Fig. 3, is present in experimental data. This would also help the reader appreciate the relation and differences between using Leonardo and the usual statistical measures.

13. The discussion about the complex potential landscape in which the particles may be moving is too qualitative. Especially, the evidence about the hierarchy of potential wells is very weak. To make it stronger, the authors should perform

an analysis similar to that shown in Fig. 5 with simulations of FBM and CTRW. They should also show the correlation between waiting times and z_{12} . Otherwise, the authors should carefully rephrase the paragraph to make it clear that they are just suggesting the landscape and not demonstrating. Similarly, the last sentence in the conclusion should be removed. To make it really quantitative, the authors might consider explicitly simulating the hierarchy of potential they discuss, although, this may be beyond the scope of this paper.

Minor comments

- In line 76, I find the choice of the pronoun “they” when referring to a generative AI odd.
 - In line 125, it is unclear what the authors mean by “within the limit of normal diffusion”.
 - In line 145, physics-informed machine learning does not only adjust the hyperparameters but affects the whole training.
 - The definition and introduction of the statistical features around line 149 could be streamlined, for example, by defining μ and σ from the start.
- In line 201, the fact that FBM and CTRW are very different mechanisms makes it easier to tell them apart, not harder, as the authors seem to suggest.
- It would be helpful to have some more details about the p-values given in lines 215, 216. What is the null hypothesis there? I think the whole discussion of Fig. 2 could be streamlined. It would also be useful to recall a definition of the absolute correlation coefficient.
- The authors claim “Beyond merely simulating known behaviors, the model has other potential applications, including the exploration of novel nanoparticle dynamics, assisting in hypothesis generation for experimental settings, and serving as a predictive tool for interpreting complex diffusive behaviors.” but do not justify the claim nor explain how they would do what they suggest.
- Typo in line 324, it should be Fig. 5 d, i.
 - Reference 18 contains a typo.

(Remarks on code availability)

Version 1:

Reviewer comments:

Reviewer #1

(Remarks to the Author)

See attached file.

(Remarks on code availability)

I thank the reviewers for including an introductory notebook to their code.

Reviewer #2

(Remarks to the Author)

See attached file.

(Remarks on code availability)

Version 2:

Reviewer comments:

Reviewer #1

(Remarks to the Author)

I thank the authors for responding to my comments and adapting their manuscript based on the raised concerns. I think the manuscript has greatly improved through the revision process and is ready for publication.

I have just some minor comments left, which may not need any further change to the manuscript.

First, I must confess that I don't like the addition of a UMAP, as I think that applying it to the latent space of a properly trained VAE shouldn't give rise to any more information. I also understand that this was added as a response to Reviewer 2.

Let me justify the previous: if the VAE has been properly trained, this means that it has reached a minimal representation of the data, through collapsing all unnecessary neurons to the prior. Hence, a UMAP or any other dimensionality reduction method cannot reduce the dimension any more. They are just going to perform a non-linear transformation of what is already encoded in the surviving neurons. One of the advantages of VAE, in my opinion, is that they avoid the use of such dimensionality reduction methods, and their representations are typically much more interpretable (as shown by this work and many others). As a curiosity, it has been shown that VAEs actually perform PCA in the latent space (see <https://arxiv.org/abs/1812.06775>).

My second minor comment relates to the values of the log 2 and the general training of the VAE. Looking at the table shown in the author's response, I think the training of the VAE can probably be improved. As the authors show throughout the paper, only two parameters are needed to describe the data: velocity autocorrelation and non-Gaussianity. I understand their comment about the network splitting the non-Gaussianity in two dimensions, although in the ideal scenario, I would expect that a single neuron should be sufficient for homogenous diffusion. What concerns me is the high values of log 2, as the authors also note in their response. I believe their high value is due to a too small regularization of the latent space (i.e. too small w_2 weight in their loss). Typically, finding the correct value for such a term is the main challenge in training VAEs, and given that the authors have a quite complex loss, it may be even harder. Perhaps of interest to the authors is the TC-VAE loss (Eq 2 in <https://arxiv.org/pdf/1802.04942>), which typically gives rise to much better disentangled representations in the latent space and is easier to hyperparameterize.

Note that, if one wants to use VAEs in a completely unknown scenario, we need to ensure that the least prior-knowledge is used in the analysis of the latent space, hence the need for models that isolate extremely well all physical variables in only the necessary latent neurons.

(Remarks on code availability)

The code provided by the authors has allowed me to test their method and is a valuable tool for researchers reading their paper.

Reviewer #2

(Remarks to the Author)

The authors have addressed most of my comments.

However, two of my comments were not discussed in sufficient depth and were ignored in the revised version of the manuscript.

The authors did not include any discussion in the manuscript about the relation between the classifiers and the Fréchet distance and their abilities to measure the similarities of the generated trajectories with the experimental dataset (my second comment). The reply to my comment in the rebuttal, though not really exhaustive, highlights some subtleties that the authors glossed over in the manuscript. I therefore invite the authors to expand this discussion and to include at least a summary of it in the revised manuscript.

Similarly, in my first comment, I asked to provide additional metrics to quantify the quality of the generated trajectories. While the FID is an established metric for studying generated images, it has not been widely used for assessing the quality of generated stochastic time series (in case I am wrong, I'd be happy to see references about this). It is therefore not sufficient to blindly take it as the only relevant metric, and additional ones are required. Even though the metrics I suggested are used for the training, it is not uncommon to show the performance of machine-learning models on a test dataset in terms of the metrics used for training. Especially, highlighting how different such metrics are with respect to those of other anomalous diffusion metrics will strengthen the claims of the authors.

(Remarks on code availability)

Version 3:

Reviewer comments:

Reviewer #1

(Remarks to the Author)

I thank the authors for answering my minor comments, and in general for considering my previous remarks in their revised versions of the text. I think the work will be a great contribution to the field, showcasing that unsupervised learning approaches can be used to discover (re-discover in the case of the paper) new physics directly from experiments.

(Remarks on code availability)

The code provided by the authors has allowed me to test their method and is a valuable tool for researchers reading their paper.

Reviewer #2

(Remarks to the Author)

The authors have addressed all my remarks and I am happy to recommend their manuscript for publication.

(Remarks on code availability)

Dear Editor,

We appreciate the reviewers' responses to our manuscript. We are submitting a revised version based on the reviewers' comments. An additional version of the paper is included where all major changes are in **blue** font. In this reply, our responses to the reviewers and the relevant sections of the revised manuscript are noted in **blue**.

Response to Reviewer #1:

Reviewer Comment: The authors present an unsupervised method to extract the physical features from the motion of nanoparticles in LPTEM experiments. In particular, they use a VAE with a physics-inspired loss function that helps the model capture the properties of input trajectories. The work tackles an important topic: the autonomous extraction of new knowledge, in terms of physical features, directly from stochastic data, in an unsupervised way. The work is very well presented and the goals and methods to achieve them are clear. The associated code is useful for the experts and helped me check some details about the dataset and model use. It is however not suitable for non-experts, who may need further information or an “introductory” notebook.

We thank the reviewer for appreciating our work and for providing helpful comments.

I have few questions regarding the technical aspects of the work. To summarize what I expand in the comments below: it is not clear which latent features have been exactly learned by the VAE. I think that: 1) the VAE could have just learned the experimental properties being changed in the experiments (beam dose rate and nanorod size); 2) the simulated dataset added to the training may have biased the VAE towards the properties of FBM and CTRW. I hope that my questions below help clarify these points.

1. Major comments:

1.1) First, I would like to point out that a recent work (Fernandez-Fernandez et al. PRE Phys. Rev. E 110, L012102 (2024)) tackles the same problem with a VAE, although without a physics informed loss. However, in that case the “reconstruction” loss (i.e. L_1 of the authors loss) is changed to a probabilistic loss that compares the distribution of the decoder’s output to the distribution of the dataset. In this way, all terms considered by the authors for their physics informed loss are taken into account implicitly.

We thank the reviewer for pointing out the recent work by Fernández-Fernández et al. (Phys. Rev. E 110, L012102 (2024)) [1] and its relevance to this study. While their use of a variational autoencoder (VAE) with a probabilistic reconstruction loss provides a solid framework for modeling Gaussian processes, we believe that this approach has limitations when applied to non-Gaussian experimental data such as the trajectories analyzed in our work.

The reconstruction loss in Fernández-Fernández et al. assumes that the data and the decoder's output can be modeled as Gaussian distributions. This is evident from the use of the negative log-likelihood under a Gaussian probability density function, which forms the core of their loss function. Specifically, their reconstruction loss function is defined as:

$$\mathcal{L}_{recon} = -\log p_{\theta}(\Delta x|z) = -\sum_{i=1}^N \sum_{t=1}^{T-1} \log N(\mu_t^{(i)}, \sigma_t^{(i)2}),$$

Where for a single displacement at time t :

$$-\log N(\mu_t, \sigma_t^2) = \log(\sigma_t \sqrt{2\pi}) + \frac{(\Delta x_t - \mu_t)^2}{2\sigma_t^2}.$$

Here, Δx_t represents the displacement at time t , μ_t and σ_t^2 are the predicted mean and variance of the decoder’s output and N is the total number of samples in the dataset. This formulation assumes that the decoder’s output for each trajectory follows a Gaussian distribution, parameterized by μ_t and σ_t^2 , at each time step.

While this approach is effective for processes with Gaussian characteristics, e.g., Brownian motion, fractional Brownian Motion (FBM), or scaled Brownian Motion (SBM), which are the three ideal diffusion process considered in their work, it is unsuitable for systems with non-Gaussian displacement distributions. Experimental data from LPTEM trajectories, for instance, often exhibit heavy tails or high skewness due to the heterogenous and viscoelastic nature of the environments around the particles.

1.2) In that same work, the authors claim that an autoregressive decoder is needed in order to generate FBM trajectories, as only in this case the decoder’s output can show the proper correlations. While the authors show that the generated show some correlation via the autocorrelation plots, to which extent their model can actually generate trajectories with long range correlations as FBM ones? I suspect that while the transformer decoder may be close to do so to some extent, the linear layer at the end of the decoder may “kill” most of the correlations.

We thank the reviewer for raising this important point regarding the decoder architecture and the ability of LEONARDO to capture long-range correlations, such as those observed in fractional Brownian motion (FBM) trajectories. In light of this, we replaced the linear layer at the end of the decoder with a convolutional layer to better preserve correlations in the generated trajectories. The convolutional layer facilitates the learning of local patterns while maintaining temporal dependencies in the trajectory data, mitigating the potential issue of the linear layer "killing" correlations.

Additionally, LEONARDO's loss function includes specific terms explicitly designed to capture velocity and positional autocorrelations. This ensures that the model learns these temporal dependencies directly from the data. Since this work focuses on short-range correlations (e.g., at $\tau = 1$), Figure 3 demonstrates that latent variable z_4 is highly correlated with the velocity autocorrelation at $\tau = 1$. This observation highlights LEONARDO's capability to model the relevant statistical properties of the trajectories, including temporal correlations.

To further address whether the model captures patterns observed in experimental data, we introduced a Frechet Inception Distance (FID)-based metric, commonly used in the generative AI community for evaluating the performance of generative models. This metric, Frechet Distance (FD) for our case, measures the similarity between the statistical distributions of experimental and generated trajectories. The FD results (shown below) show that the distance between the generated and experimental data is very low, confirming that LEONARDO successfully captures the intricate patterns and properties of real LPTEM trajectories.

Together, the architectural changes (addition of a convolutional layer as suggested by the reviewer), the inclusion of correlation terms in the loss function, and the strong performance metrics (FD and classification accuracy) provide solid evidence that LEONARDO can generate trajectories with meaningful correlations,

even for complex processes such as those observed in experimental LPTEM data. Detailed discussions on the FD metric and classification performance can be found in section 2.2 of the revised Manuscript.

1.3) About the dataset, I fail to understand the purpose of augmenting it with simulated data. If the authors aim to extract the physical features of their experiment, wouldn't it be enough to train with experimental trajectories? Following the author's train of thought, if we are not sure about the true model describing the experiments, can not the simulated dataset wrongly bias the VAE latent? Have the authors trained a VAE purely with experimental data? What latent space would arise from there?

We thank the reviewer for this comment regarding the inclusion of simulated data. We agree that using simulated data could potentially bias the latent space representation of LEONARDO if the simulated data does not perfectly reflect the dynamics of the experimental trajectories. In response to this concern, we have revised our approach to focus exclusively on experimental data.

To achieve this, we collected an additional ~17,000 experimental trajectories to add to the training dataset, covering a range of electron beam dose rates and particle sizes. This significantly expanded dataset allowed us to train LEONARDO without the need for simulated data. Furthermore, we transitioned from analyzing 1D trajectories to 2D trajectories, as suggested by both reviewers, further emphasizing our commitment to grounding the model in experimentally observed dynamics.

By training LEONARDO solely on experimental data, we make no assumptions about the underlying nature of the trajectories, such as their similarity to a combination of fractional Brownian motion (FBM) and continuous-time random walk (CTRW), as we had previously done. This adjustment ensures that the latent space learned by LEONARDO is informed entirely by the properties and statistical features of real LPTEM data and that the results and conclusions drawn from our analysis are fully representative of the experimental system under investigation.

1.4) Related to the latter, I wonder what is exactly in the latent space of the VAE. The ML literature has shown us that these models are able to learn the factors of variations of a given dataset.

A) If focusing only on the experimental dataset, there are two factors of variation: the size of the nanorod and the beam dose rates. I am not an expert on this experiment, but I understand that these properties will affect the diffusion coefficient and the anomalous exponent. I understand that the diffusion coefficient here is not a factor of variation due to the normalization procedure.

B) If considering that the simulated dataset has exactly the same properties as the experimental dataset, we have at least three factors of variation: 1) the value κ of Eq. S5; 2) the α of the CTRW; 3) the α of the FBM. Actually, are these two latter sample independently? I also think that in this case the generalized diffusion coefficient should enter into play as the normalization procedure may affect FBM and CTRW trajectories differently, as D_{α} depends on α in two different ways.

Both these cases make me wonder what exactly has the VAE learned? Is the VAE extracting the experimental features (size and dose) or the physical features (κ s, α s, D_{α} s). I think some ablations on the dataset (i.e. training only on experiments / simulated) may help discern this.

We thank the reviewer for this thoughtful question about the factors of variation learned by LEONARDO and the insights provided by the latent space. As described in Section 2.3 in our revised manuscript, LEONARDO's latent variables learn to encode statistical properties of the trajectories as defined by the physics-informed loss function. Specifically, latent variables z_1 , z_4 , and z_5 demonstrate strong correlations with non-Gaussianity and velocity autocorrelation, which are key statistical indicators suggestive of the heterogeneity and viscoelasticity of the interaction energy landscape surrounding nanoparticles (Figure 3). These latent variables allow LEONARDO to capture meaningful patterns in the experimental data.

To address the reviewer's specific query about the experimental factors of variation, such as particle size and beam dose rate, we expanded our previous Section 2.4 analysis to include the effect of particle size in addition to the electron beam dose rate study that was already part of our work. Based on the reviewer's suggestion, we recognized that particle size is an important factor of variation in the dataset and incorporated it into our analysis. To do this analysis, we conducted additional experiments to collect trajectories for nanoparticles with a size of 40 nm under dose rate of $35 e^{-}/\text{\AA}^2 \cdot s$. As described in the revised Section 2.4, LEONARDO effectively distinguishes between these experimental conditions, as shown in our analysis of latent space distributions in Figure 4 (shown below). For example, the latent variables z_1 and z_5 , which are associated with non-Gaussianity, exhibit distinct distributions for different dose rates and particle sizes, as shown in Figures 4a and 4b. Similarly, z_4 , which correlates strongly with velocity autocorrelation, captures differences in the viscoelasticity of the environment across these experimental conditions (Figures 4e and 4f). These results highlight the model's ability to extract physical features directly tied to the experimental parameters. However, it is important to note that the effects of particle size and dose rate are not entirely independent in this system. Unlike simpler cases where size-dependent factors such as the diffusion coefficient scale predictably with particle size, the diffusion behavior in LPTM is more complex. In this environment, the interactions between nanoparticles and the silicon nitride membrane are influenced by both dose rate and size, leading to overlapping effects on the trajectories.

Regarding the potential influence of simulated data, we have completely removed simulated trajectories from our training dataset in the revised manuscript, as noted in our response to Reviewer Comment 1.3. LEONARDO is now trained solely on experimental LPTM trajectories, ensuring that the latent space is shaped entirely by real-world observations. This revision eliminates any potential bias arising from assumptions about the underlying diffusion models (e.g., FBM or CTRW) and ensures that the latent space captures experimental features such as size and dose rate, as well as statistical properties.

By focusing solely on experimental data, the latent space learned by LEONARDO reflects the statistical features that vary naturally in the experimental system, rather than any predefined or simulated parameters such as κ , α_{FBM} , or α_{CTRW} . This is further supported by our discussion in our revised Section 2.3, where we show that the latent variables capture non-Gaussianity and velocity autocorrelation, without being explicitly parameterized by diffusion coefficients or exponents.

Fig. 4 Characterization of the interaction energy landscape in LPTEM using LEONARDO. **a**, Probability densities of the values of $\max(z_1^2, z_5^2)$ for experimental trajectories collected at electron beam dose rates of 20 and 30 $e^-/\text{\AA}^2\cdot\text{s}$ and particle size of 60 nm. **b**, Probability densities of the values of $\max(z_1^2, z_5^2)$ for experimental trajectories collected at particle sizes of 40 nm and 60 nm at an electron beam dose rate of 35 $e^-/\text{\AA}^2\cdot\text{s}$. **c**, Probability densities of the values of z_4 for the same experimental trajectories as in (a). **d**, Probability densities of the values of z_4 for the same experimental trajectories as in (b). **e**, Probability densities of the values of non-Gaussianity at $\tau = 1$ for the same experimental trajectories as in (a) and (c). **f**, Probability densities of the values of non-Gaussianity at $\tau = 1$ for the same experimental trajectories as in (b) and (d). **g**, Probability densities of the values of velocity autocorrelation at $\tau = 1$ for the same experimental trajectories as in (a), (c), and (e). **h**, Probability densities of the values of velocity autocorrelation at $\tau = 1$ for the same experimental trajectories as in (b), (d), and (f).

1.5) A common feature of VAEs is that the variance of most of the neurons will go to 1 (as given by the prior) while the neurons needed by the decoder to generate new data converge to something smaller. Have the authors checked this? In their case, only the neurons 2 and 12 should have small variances while the rest should go to 1.

We thank the reviewer for this insightful comment regarding the variances of latent variables and their connection to the features learned by LEONARDO. The observation that variances of latent variables in VAEs converge to 1, with only the relevant latent variables having smaller variances, was mentioned in previous literature [1], where the authors note that meaningful latent neurons exhibit smaller variances when they encode key characteristics of the data.

To address the reviewer’s comment, we performed an additional example analysis to investigate the variances of the latent variables in LEONARDO with statistical properties of the trajectories. Specifically, we calculated the variance of each latent variable for trajectories grouped based on their velocity autocorrelation at $\tau = 1$. The results of this analysis, summarized in the table below, show that latent variable z_4 exhibits a much lower variance (0.21) compared to other latent variables, whose variances are closer to or exceed 1.

Latent variable	Variance
z_1	0.83
z_2	0.91
z_3	0.86
z_4	0.21

z_5	0.57
z_6	0.70
z_7	0.99
z_8	1.29
z_9	0.97
z_{10}	0.74
z_{11}	0.96
z_{12}	0.91

This aligns with our findings in the revised Figure 3a, which demonstrates a strong linear correlation between z_4 and velocity autocorrelation. The low variance of z_4 indicates that it specifically captures the variations in velocity autocorrelation across the trajectories, while other latent variables do not.

However, while z_4 encodes velocity autocorrelation with a linear relationship, other statistical properties, such as non-Gaussianity, exhibit quadratic relationships with latent variables, as shown in Figure 3c. For example, z_1 and z_5 capture non-Gaussianity in the y and x -components of the displacement distribution, respectively, through a quadratic trend. As a result, the same analysis on these latent variables with non-Gaussianity would not show a low variance – for example, a non-Gaussianity of 6 in the y -component can be achieved with latent variable $z_1 \approx -3$ or $z_1 \approx 3$.

Therefore, while we acknowledge the importance of this additional analysis, we believe that the revised Figure 3a addresses the core of the reviewer’s concern. The correlations in Figure 3a (shown below) clearly show which latent variables are learning meaningful statistical properties, such as non-Gaussianity and velocity autocorrelation, and which are not. This figure provides a direct and comprehensive view of the relationships between latent variables and key trajectory features, making it sufficient to demonstrate the relevance of each latent variable. We thank the reviewer for raising this point, as it allowed us to further validate our approach and clarify the relationships captured by the latent space.

R^2 x-component $\xi(\tau=1)$	0.04	0.00	0.25	0.20	0.87	0.01	0.00	0.02	0.00	0.02	0.00	0.00
R^2 y-component $\xi(\tau=1)$	0.79	0.00	0.32	0.30	0.06	0.00	0.00	0.01	0.00	0.01	0.01	0.00
R^2 x-component $C_v(\tau=1)$	0.01	0.00	0.08	0.90	0.04	0.00	0.00	0.20	0.00	0.09	0.00	0.00
R^2 y-component $C_v(\tau=1)$	0.08	0.00	0.02	0.93	0.00	0.00	0.00	0.10	0.00	0.09	0.00	0.00
	1	2	3	4	5	6	7	8	9	10	11	12
	Latent variable, z_i											

1.6) Given the previous, all neurons but 2 and 12 are meaningless, as their variance is very big. This means that the effective latent manifold is of dimension 2. Hence, why is the TSNE in Figure 2 needed? Could not a similar plot be achieved by plotting z_2 vs z_{12} ?

We thank the reviewer for their correct observation regarding the dimensionality of the latent space and the t-SNE plot in the previous version of the manuscript. In the revised manuscript, we have removed the t-SNE analysis and the corresponding discussion (previously Figure 2 and Section 2.2) because they were focused on simulated FBM and CTRW trajectories. This analysis is no longer relevant to the current study, as we have excluded simulated data from the training dataset as noted in our response to Reviewer Comment 1.4. Instead, our work now focuses exclusively on experimental LPTEM trajectories.

By removing the simulated data, we no longer make assumptions about the underlying mechanisms of the trajectories (e.g., FBM or CTRW) or their associated anomalous exponents. Instead, the latent space learned by LEONARDO now represents the statistical properties of the experimental trajectories, as defined by the terms in the loss function. This shift in focus eliminates the need for an explicit analysis of anomalous exponents or comparisons between simulated and experimental data.

1.7) While Fig.2b gives vital information to understand the latent space of the VAE, due to the number of points plotted and the colors for FBM and CTRW being so close to each other, it is hard to understand in detail what is happening. Maybe plotting the mean z will show the same in a clearer manner?

We thank the reviewer for their suggestion regarding improving the visualization of scatter plots by plotting mean values or lines of best fit to enhance clarity. While Figure 2b from the previous version of the manuscript has been removed in this revision, as it was part of the analysis focused on simulated data (now excluded from the study), we incorporated the reviewer's suggestion into our new Figure 3.

In Figure 3 (shown below), we present scatter plots of latent variables versus statistical properties, such as non-Gaussianity and velocity autocorrelation. To improve clarity and convey trends in the data, we have included lines of best fit for these scatter plots. Specifically, we used second-degree polynomial fits to capture the relationships between the latent variables and statistical properties. This approach provides a more intuitive visualization of the correlations and enhances the interpretability of the results, as shown in Figures 3c, 3d, 3f, 3g, 3i, and 3j.

We believe that this revision aligns with the reviewer's suggestion and significantly improves the presentation of data, ensuring that the relationships between the latent variables and statistical properties are more clearly conveyed.

2. Minor comments:

2.1) The authors reference [39] as an unsupervised method but that method is supervised.

We thank the reviewer for pointing out this mistake, we have made the correction in the revised manuscript.

2.2) Given that the authors have a Transformer encoder, wouldn't it be possible to feed trajectories of all sizes, if not considering the current input Linear layer? This would be an upgrade to the model to avoid having to "cut" the trajectories (and I understand may fall out of the scope of this work)

We thank the reviewer for this insightful suggestion. Our work specifically focuses on analyzing short trajectories. We want to emphasize that LPTEM experiments are very different compared to other microscopy modalities. The videos collected from one single experiment (one video in TEM microscope format) can be gigabytes of data, and because of that, users usually acquire only a limited number of trajectories. There is also a tradeoff between exposure (frames per second) and the noise level of TEM images, which pushes the users to collect videos at low frame rates (i.e., high exposure for better contrast), leaving them often with a handful of very short trajectories. So, while we collected a lot of trajectories for training this model, this is very uncommon for LPTEM experiments (considering also the cost of the experiments). For this reason, we chose to work with fixed-length trajectories of 200 points (which can correspond to a 200-second video at 1 frame per second). Additionally, while it is theoretically possible to adapt the Transformer encoder to handle trajectories of variable lengths, this would introduce additional challenges. Allowing for variable-sized trajectories (e.g., ranging from 200 to 10,000 points) would require the model to generalize across a much broader range of sequence lengths, significantly increasing the complexity of the learning task. To achieve this, a much larger dataset would be necessary, both to capture the variability in trajectory lengths and to prevent the model from overfitting to specific ranges. Collecting such a dataset would require a substantial number of additional experiments, which is not only time-intensive but also costly given the resources involved in our experimental setup. Therefore, while this is an interesting direction for future work, we have focused on short, fixed-length trajectories in this study.

2.3) If I understand correctly, the authors separated the two components of the trajectory to effectively create two trajectories. Wouldn't the model benefit from having as input the two coordinates as input (e.g. a tensor of size (2×200))?

We agree with the reviewer on the benefit of considering 2D trajectories. Based on this suggestion, we revised our approach to incorporate 2D trajectories as input (tensors of size (2×200)). Since the trajectories are naturally obtained in 2D, it is appropriate to analyze them directly in this form, as this preserves the correlations between the x and y components that are otherwise lost when splitting the data into 1D trajectories.

Initially, when using 1D trajectories, we had a larger dataset because each trajectory was split into its x and y components, effectively doubling the data size. To ensure a comparable dataset size in the 2D case, we conducted additional experiments to collect sufficient data. We have also mentioned this in our response to Reviewer Comment 1.3. This change also resulted in modifying all our loss function terms to account for 2D terms by considering the position vector $r = (x, y)$.

2.4) There seems to be a typo in line 143: traction -> attraction?

The term "traction" was used intentionally to convey that physics-informed machine learning has gained momentum and widespread adoption in the field.

To summarize, I think the VAE approach shows great potential for the study of anomalous diffusion. The authors make a great effort to apply it to experimental data. However, because they combine the training

with simulated data, it is hard to dissociate exactly what has been learned in the latent space. I hope some of my concerns help clarify these points.

Reviewer #1 (Remarks on code availability):

The associated code is useful for the experts and helped me check some details about the dataset and model use. It is however not suitable for non-experts, who may need further information or an “introductory” notebook.

We agree with the reviewer on the importance of code accessibility. In response to this comment, we have added an introductory notebook to the project website. This notebook provides step-by-step instructions for setting up and running the code, exploring the latent space, and generating trajectories from LEONARDO. It is designed to guide non-expert users in understanding and utilizing the code effectively, ensuring that it is accessible to a broader audience. We hope this addition enhances the usability of our work for researchers at varying levels of expertise.

Summary of responses to Reviewer 1 comments

We sincerely thank the reviewer for their detailed feedback, which has been instrumental in improving the quality of our manuscript. Below, we summarize our responses to the key points raised:

1. **Clarification of the latent space and dataset:**

We addressed the concern regarding the latent space learning properties of the VAE by focusing entirely on experimental data in this revised manuscript. Simulated data, previously included in our training, has been removed to ensure that LEONARDO's latent space is shaped solely by real-world experimental features. To achieve this, we collected additional experimental trajectories at varying electron beam dose rates and particle sizes. As a result, our analysis now clearly demonstrates how LEONARDO learns statistical properties (e.g., non-Gaussianity and velocity autocorrelation) as defined by the physics-informed loss function (Section 2.3). Furthermore, we incorporated the reviewer's suggestion to analyze particle size as a factor of variation alongside dose rate, as presented in Section 2.4.

2. **Decoder architecture and ability to capture correlations:**

In response to concerns about preserving trajectory correlations, we replaced the linear layer at the decoder's output with a convolutional layer. Additionally, the physics-informed loss function explicitly incorporates velocity and positional autocorrelations, ensuring that these temporal dependencies are directly learned by the model. Performance metrics, such as the Frechet Distance (FD), further validate LEONARDO's ability to generate trajectories resembling experimental data. The FD results show a small distance between experimental and generated trajectories, confirming the model's success in capturing key statistical patterns.

3. **Improved visualizations:**

We adopted the reviewer's suggestion to enhance scatter plot clarity by including best-fit polynomial lines in the analysis of latent variables and statistical properties (Figure 3). This ensures that trends in the data are more apparent and the relationships between latent variables and trajectory features are conveyed effectively.

4. **2D trajectories and expanded dataset:**

Following the reviewer's suggestion, we revised our approach to analyze trajectories in their

native 2D form rather than splitting them into independent x and y components. This preserves inter-component correlations and aligns with the natural structure of the data. To achieve a comparable dataset size, we collected additional experimental trajectories. This revision further strengthens the analysis and ensures that LEONARDO operates on data representative of the experimental system.

5. Code availability and accessibility:

In response to the reviewer's comment about the accessibility of the code for non-experts, we have included an introductory notebook on the new project website. This notebook provides step-by-step instructions and explanations for setting up and using the code, making it more approachable for users at varying levels of expertise.

We appreciate the reviewer's recognition of the potential of the VAE approach for studying anomalous diffusion and believe that the changes made significantly strengthen the work.

Response to Reviewer #2 :

Reviewer Comment: The authors developed an interesting method to generate stochastic trajectories via a neural network architecture. The generated trajectories match some of the statistical properties of the ones the networks were trained on but are new, independent trajectories. The statistical properties chosen by the authors are the mean, variance, skewness, kurtosis of the displacements and the particle velocity autocorrelation. The neural network architecture involves an encoding part that compresses the input trajectory and encodes their salient features, followed by a decoding part that generates new trajectories possessing those encoded features. Interestingly, by inspecting the latent space where the features are encoded, it is possible to gain some physical insight into the input trajectories and to generate trajectories with similar characteristics. I find this approach interesting, and the attempt to increase the interpretability of neural networks for single particle trajectories is a much-needed and ambitious effort. However, I feel the manuscript provides mostly qualitative evidence, thereby falling short of fully reaching its goal of providing an interpretable machine-learning method. It also contains many bold claims that are only loosely motivated. I will provide my detailed criticism below.

We thank the reviewer for appreciating the work and for the insightful comments.

MAIN COMMENTS

1. The authors do not provide convincing metrics about the performances of their method. It is then impossible for the reader to understand how “good” these models are at generating or analysing trajectories. How closely do the generated trajectories capture the experimental ones? (see my comment below as well).

We thank the reviewer for the helpful comment, which led us to use two performance metrics to assess how closely the LEONARDO-generated trajectories resemble the experimental trajectories in the LPTM dataset.

Performance Metric 1: Fréchet Distance between learned feature vectors

The first performance metric we used is based on the Fréchet Inception Distance (FID). The FID is a widely recognized metric introduced by Heusel et al. (2017) [2] for assessing the quality of generative models. While originally designed for evaluating generative models trained on images, the concept of FID is equally applicable to other domains, where the objective is to measure the similarity between generated and real datasets in a learned feature space.

The FID measures the Fréchet distance (also known as the Wasserstein-2 distance) between two distributions: one representing features of the real data and the other representing features of the generated data. These distributions are modeled as multivariate Gaussians. To compute the FID for image data, Heusel et al. utilized features extracted from a pretrained Inception-v3 classification network. The features were taken from the output of the second-last layer (before the dense layer that maps to the number of classes), which provides a high-level representation of the input data. This design ensures that the FID score captures semantic differences between images rather than pixel-level discrepancies. FID has been shown to correlate well with human judgment of image quality, making it a reliable measure for evaluating image generation models. The FID calculates the Fréchet distance, d , between two multivariate Gaussian distributions P_r and P_g , using the following formula:

$$d^2 = \|\mu_r - \mu_g\|_2^2 + \text{Tr} \left(\Sigma_r + \Sigma_g - 2(\Sigma_r \Sigma_g)^{\frac{1}{2}} \right),$$

where μ_r, Σ_r and μ_g, Σ_g are the mean and covariance matrices of the features extracted from the real and generated data. A lower FID score indicates greater similarity between the two distributions.

While the Inception-v3 model is highly effective for image data, it cannot be used for trajectory data like the LPTEM dataset. Therefore, we adapted this Fréchet distance notion for time-series data as described in the following. We used MoNet, our previously published deep learning-based model for the classification and characterization of anomalous diffusion processes[3]. MoNet leverages a convolutional neural network architecture to process input trajectories and extract features that distinguish between different diffusion classes. We expanded the applicability of MoNet by training it on new classes: LPTEM trajectories, simulated Brownian Motion (BM), and simulated anomalous diffusion trajectories from the Anomalous Diffusion (AnDi) Challenge models [4], which provides a standardized dataset and framework for generating anomalous diffusion trajectories. The AnDi trajectories used to train MoNet were Fractional Brownian Motion (FBM), Continuous-Time Random Walk (CTRW), Annealed Transient Time Motion (ATTM), Scaled Brownian Motion (SBM) and Levy Walk (LW). We used 8000 2D trajectories from each diffusion class. The confusion matrix below shows the classification performance of the MoNet classifier on test data. It achieved an overall weighted F1 score of 0.88.

Fréchet distance between diffusion classes

After training MoNet, we extracted its second-last layer output (a 128-dimensional feature vector) for both the LPTEM trajectories and the LEONARDO-generated trajectories. This layer is a high-level feature representation of trajectories, analogous to the second-last layer of Inception-v3 for images. Using these feature vectors, we calculated the Fréchet distance between the distributions of the LPTEM trajectories and the LEONARDO-generated trajectories which is 0.547. For reference, scores between different diffusion classes ranged between ~ 1.1 and ~ 2.8 . The scores between different diffusion classes are visualized below in the form of a lower triangular matrix, where each cell represents the FD score between the feature

distributions of two diffusion classes (e.g., FBM vs. LPTM, CTRW vs. LPTM, etc.). A low score between the LEONARDO-generated trajectories and LPTM trajectories indicates that LEONARDO is capable of generating trajectories that are highly similar to the experimental LPTM trajectories.

Performance Metric 2: Classification of generated trajectories

The second performance metric is based on "fooling" the trained MoNet classifier. As described previously, MoNet was trained to distinguish between different diffusion classes, including the LPTM trajectories. Its high F1 score on the test set demonstrates that it differentiates well between the diffusion classes. After training the MoNet classifier, we evaluated its classification of LEONARDO-generated trajectories. If the LEONARDO-generated trajectories resemble LPTM trajectories more than they resemble other diffusion classes, we expect the classifier to predominantly label them as LPTM rather than as other diffusion types. For this evaluation, we input LEONARDO-generated trajectories into MoNet and recorded the predicted diffusion types. The results, summarized in a bar chart below, show that over 95% of LEONARDO-generated trajectories are classified as LPTM trajectories, highlighting their strong similarity to LPTM trajectories.

How reliable and accurate is the quantitative inference of the anomalous diffusion exponent contained in the latent variables? Many methods have been developed to estimate the anomalous exponent both using traditional [e.g., V. Tejedor, et al, Biophys. J. 98, 1364 (2010) K. Burnecki, et al, Sci. Rep. 5, 11306 (2015). Y. Meroz and I. M. Sokolov, Phys. Rep. 573, 1 (2015). N. Makarava, et al, Phys. Rev. E 84, 021109 (2011). A. Weron, et al Phys. Rev. E 99, 042149 (2019), J. H. Jeon and R. Metzler, J. Phys. A: Math. Theor. 43, 252001 (2010), Krapf, and M. M. Tamkun, Biophys. J. 111, 1235 (2016), G. Sikora, et al, Phys. Rev. E 96, 062404 (2017). A. Weron, et al, Sci. Rep. 7, 5404 (2017)] and AI tools. The authors should show how their method performs compared to them, for instance by using the AnDi dataset [Muñoz-Gil, G. et al. Nature Communications 12, 6253 (2021)], which provides an excellent benchmark.

We thank the reviewer for their comment. In the revised submission, we have removed Figure 2 from the original submission along with its accompanying analysis based on feedback from Reviewer 1. This decision results from a key change in our revised approach: excluding simulated trajectories from the LEONARDO training dataset to train it exclusively on LPTM trajectories. By making this adjustment, we eliminated the assumption that the latent space learns important features of FBM and CTRW trajectories. Consequently, analyzing their alpha exponents was no longer relevant, and we deemed it appropriate to omit this analysis.

2. The authors should justify why they chose the specific 5 statistical features mean, variance, skewness, kurtosis of the displacements and the particle velocity autocorrelation to assess the similarity of the trajectories. For example, the authors of A Gentili and G Volpe 2021 J. Phys. A: Math. Theor. 54 314003 used many more features (over 90) to properly assess the anomalous diffusive behaviour of single trajectories. How can the authors confidently claim that trajectories sharing their chosen 5 properties are effectively representing the same process, without resorting to more refined spectral properties of the signal and correlations? This analysis is necessary to confidently show that the trajectories are indeed accurate and realistic. Interesting works that could be helpful in this direction are, for instance, Krapf New J. Phys. 20 (2018) 023029, which analysed what can be learned from the power spectral density of a single Brownian trajectory and analysed anomalous diffusion in Krapf Phys. Rev. X 9, 011019

We thank the reviewer for their comment regarding the choice of statistical features used to represent the LPTM trajectories. We agree that this is a crucial aspect to address, and it has prompted us to carefully evaluate the impact of additional statistical features on the performance of LEONARDO using an ablation study. Below we have included a detailed analysis of each loss term. The Appendix at the end of this letter includes all figures associated with this analysis.

In our baseline version of LEONARDO, the model was trained using mostly the same terms as in the original submission, specifically the first four moments of the displacements: mean, variance, skewness, and kurtosis, and velocity autocorrelation terms. To adapt to 2D trajectories, we added an additional term to account for x-y correlations. This baseline LEONARDO model served as the foundation for the ablation study we conducted, allowing us to systematically examine how incorporating additional statistical features impacts model performance as measured by the two aforementioned performance metrics (please see response to comment 1). The statistical properties we considered in this study were drawn from the extensive set of 92D (where D is the dimension of the trajectories) statistical features provided in the CONDOR framework [5], which characterizes anomalous diffusion trajectories. This framework offers a rich set of features designed to capture different aspects of trajectory dynamics, including displacements, spectral properties, and time-series asymmetries.

The first four statistical properties included in the CONDOR framework are the moments of the displacements: mean, variance, skewness, and kurtosis. These are the same features we used in the baseline LEONARDO model. Building on this, we evaluated additional features inspired by the CONDOR paper to determine their impact on LEONARDO's performance. Below, we provide a list of the statistical properties considered in our investigation. For a detailed description of each property, we refer readers to the CONDOR paper. It is worth noting that where the CONDOR framework uses "standard deviation," we use "variance" for consistency with our existing features.

1. Median of displacements of trajectories
2. Mean, median, variance, skewness, and kurtosis of the absolute values of the normalized displacements.
3. Mean, median, variance, skewness, and kurtosis of the eight vectors of the absolute values of displacements taken at eight different time lags.
4. Median of the ratios between consecutive normalized displacements.
5. Mean, median, variance, skewness, and kurtosis of the absolute values of the discrete Fourier Transform of the displacements centered at the zero-frequency component and normalized to the trajectory length.
6. Mean, median, variance, skewness, and kurtosis of the integrated power spectral density (IPSD) as described in [5].

7. Mean, median, variance, skewness, and kurtosis of the absolute values of the variations of the time-averaged mean squared displacement normalized to the corresponding time lag.
8. Mean, median, variance, skewness, and kurtosis of the absolute values of wavelet coefficients using the Morlet wavelet function.
 - The authors of the CONDOR framework used the built-in MATLAB function *cwt* with its default settings to calculate the continuous wavelet transform (CWT) of the displacement data, specifically employing the analytic Morse wavelet. To replicate this process in Python, we implemented the CWT using the Morlet wavelet, which is comparable in purpose and functionality to the analytic Morse wavelet. The CWT was performed on the displacement data at two scales ($s = [1,2]$) and we computed the absolute values of the wavelet coefficients corresponding to the first two translations. From these coefficients, we extracted the statistical moments (mean, variance, skewness, kurtosis, and median).
9. The integral IACF as described in [5].
10. The level of frequency asymmetry as described in [5].
11. The level of time asymmetry as described in [5].
12. Absence of abrupt changes as described in [5].

Please note that we did not include Approximate Entropy (AppEn) of each time series, another feature used in the CONDOR framework, despite its potential to provide valuable insights into time-series regularity. AppEn involves comparing pairs of vectors in a time series to determine the likelihood of similarity at varying time scales. This process includes steps like thresholding and counting, which result in a non-differentiable function. In the CONDOR framework, AppEn was precomputed and used as input to the model, meaning backpropagation was not performed over AppEn or any of the statistical features. However, in LEONARDO, where the loss function directly incorporates statistical features in the loss function, the non-differentiable nature of AppEn makes it incompatible with backpropagation, which relies on smooth, differentiable functions to compute gradients and update model parameters.

In addition to the statistical properties used in the CONDOR framework mentioned above, we also evaluated the positional autocorrelation function of trajectories as a component of the loss function. The positional autocorrelation function, defined as $C_r(\tau)$ below, quantifies the correlation between the particle's position at two different time points separated by time lag τ .

$$C_r(\tau) = \frac{\langle \mathbf{r}(t) \cdot \mathbf{r}(t + \tau) \rangle}{\langle \mathbf{r}^2(t) \rangle},$$

where $\mathbf{r}(t)$ and $\mathbf{r}(t + \tau)$ are the position vectors of the particle at time t and $t + \tau$ respectively, $\langle \cdot \rangle$ denotes the average value and the denominator normalizes the positional autocorrelation by the variance of the position. We deemed this feature to be particularly important for LPTEM trajectories because these trajectories often exhibit distinct behaviors where particles abruptly jump from one position to another and then remain localized at the new position. This behavior introduces long-term positional correlations that are not fully captured by statistical properties of the displacements, such as their moments or autocorrelations. Incorporating the positional autocorrelation function as a loss term ensures that LEONARDO accounts for these spatial dependencies and better reflects the dynamics observed in LPTEM experiments.

Before carrying out the ablation study, we first systematically assessed whether the baseline model already accounted for the statistical features listed above. To achieve this, we calculated the losses associated with all the statistical features over input and generated trajectories. The objective was to identify which statistical features, if any, were implicitly learned by the baseline model without requiring explicit inclusion in the loss function. For this investigation, we retrained the baseline model over 1, 20, 40, 80, 120, and 160 epochs, creating six “intermediate baseline models”. For each statistical feature, we tracked how its associated loss evolved across these intermediate models. If the loss for a specific feature consistently decreased over successive intermediate models, it was interpreted as evidence that the baseline model inherently captured that feature. Conversely, if the loss remained constant or increased, it suggested that the feature was not being sufficiently learned and would need explicit inclusion in the loss function to ensure its representation. The figures in the Appendix section of this letter illustrate the evolution of losses over the baseline model for each of the statistical features, providing a detailed breakdown of which features required explicit consideration in the model’s training process.

Based on the evolution of losses for each statistical feature detailed in the Appendix, we identified the following statistical features as requiring explicit incorporation into the loss function of LEONARDO to investigate their impact on model performance through an ablation study. This list contains all the features for which the loss in the baseline model did not decrease over successive intermediate models.

1. Median of displacements of trajectories
4. Median of the ratios between consecutive normalized displacements.
5. Mean, median, variance, skewness, and kurtosis of the absolute values of the discrete Fourier Transform of the displacements centered at the zero-frequency component and normalized to the trajectory length.
6. Mean, median, variance, skewness, and kurtosis of the integrated power spectral density (IPSD) as described in [5].
7. Mean, median, variance, skewness, and kurtosis of the absolute values of the variations of the time-averaged mean squared displacement normalized to the corresponding time lag.
8. Mean, median, variance, skewness, and kurtosis of the absolute values of wavelet coefficients using the Morlet wavelet function.
13. Positional autocorrelation function

For the ablation study, the features identified above were incorporated into LEONARDO’s loss function one at a time, with each term weighted according to its first-epoch loss magnitude, as was done for the original loss terms of the baseline LEONARDO model. After adding each feature, LEONARDO was retrained, and its performance was assessed using the two established performance metrics (please see the response to comment 1). This approach allowed us to isolate the contribution of each feature to the overall model performance. The table below summarizes the results of the full investigation including the ablation study, highlighting the influence of each statistical feature on improving LEONARDO model performance.

Statistical features	Evolution of loss over intermediate baseline models	Ablation study	
		Change in Frechet distance from baseline model	Change in % generated trajectories classified as LPTEM, from baseline model

1. Median of displacements	Increase	-0.058	+4.39
2. Mean, median, variance, skewness, and kurtosis of the absolute values of the normalized displacements	Decrease, decrease, decrease, decrease	Not included in ablation study	Not included in ablation study
3. Mean, median, variance, skewness, and kurtosis of the eight vectors of the absolute values of displacements taken at eight different time lags	Decrease, decrease, decrease, decrease	Not included in ablation study	Not included in ablation study
4. Median of the ratios between consecutive normalized displacements	Decrease	Not included in ablation study	Not included in ablation study
5. Mean, median, variance, skewness, and kurtosis of the absolute values of the discrete Fourier Transform of the displacements centered at the zero-frequency component and normalized to the trajectory length	Decrease, decrease, decrease, increase, increase	+0.027	-0.61%
6. Mean, median, variance, skewness, and kurtosis of the integrated power spectral density (IPSD) as described in [5]	Decrease, decrease, decrease, increase, increase	+0.122	-3.01%
7. Mean, median, variance, skewness, and kurtosis of the absolute values of the variations of the time-averaged mean squared displacement normalized to the corresponding time lag	Decrease, increase, decrease, decrease, decrease	+0.036	-21.06%
8. Mean, median, variance, skewness, and kurtosis of the absolute values of wavelet coefficients using the Morlet wavelet function	Decrease, increase, decrease, increase, increase	-0.134	-10.45%
9. The integral IACF as described in [5]	Decrease	Not included in ablation study	Not included in ablation study
10. The level of frequency asymmetry as described in [5]	Decrease	Not included in ablation study	Not included in ablation study
11. The level of time asymmetry as described in [5]	Decrease	Not included in ablation study	Not included in ablation study
12. Absence of abrupt changes as described in [5]	Decrease	Not included in ablation study	Not included in ablation study
13. Positional autocorrelation function	Increase	-0.196	+16.57%

Based on this ablation study, we concluded that the positional autocorrelation function, and the median of displacements were the two features that improved both performance metrics (decreased the value of performance metric 1 and increased the value of performance metric 2), and therefore, we decided to include these in the loss function of LEONARDO. In contrast, the inclusion of other statistical terms resulted in a

decrease in at least one of the two model performance metrics. This decrease is likely due to the redistribution of relative weights in the loss function – adding additional terms reduces the proportional weight of the original terms. These findings suggest that, relative to the terms already present in the baseline model, the additional features do not provide as significant a contribution to improving the two performance metrics. Thus, the baseline features, along with the two newly added terms, capture the key statistical properties required for optimal performance of LEONARDO.

3. The authors state: “The short single-particle trajectories used to train Leonardo allow the analysis of any length of real experimental data by simply segmenting the longer experimental trajectories into shorter pieces and analyzing their time-evolving behavior”. Very often, learning from short trajectories returns methods that are best suited for short trajectories and perform poorly on longer ones. The present method may suffer from the same shortcomings. The authors should comment on this issue.

We thank the reviewer for raising this important point. To clarify, our intent was not to suggest that segmenting longer trajectories into shorter pieces would capture long-range statistics. Instead, our approach allows the analysis of longer trajectories by treating them as a series of smaller, fixed-length segments (200 time points), which are the "units" used by our VAE model. This segmentation facilitates the study of time-evolving behavior over short time scales.

For cases where longer-range statistics are of interest, a complementary approach is to subsample longer trajectories before feeding into LEONARDO, similar to the approach used when calculating time-averaged mean squared displacement (MSD) as a function of lag time (τ). By reducing the sampling frequency, one can retain information about behavior over extended time scales. To improve clarity, we have revised the relevant statement in section 2.1 of the revised manuscript as follows:

“Longer experimental trajectories can be segmented into shorter, fixed-length pieces to capture local dynamics, while subsampling of trajectories can be used to study longer-range statistics with LEONARDO.”

4. Why did the authors choose to focus on trajectories containing 200 frames?

We thank the reviewer for raising the point about the choice of trajectory length. The choice of 200 frames was guided by both experimental and computational considerations. We want to emphasize that LPTM experiments are very different compared to other microscopy modalities. The videos collected from one single experiment (one video in TEM microscope format) can be gigabytes of data, and because of that, users usually acquire only a limited number of trajectories. There is also a tradeoff between exposure (frames per second) and the noise level of TEM images which pushes the users to collect videos at low frame rates (i.e., high exposure for better contrast), leaving them often with handful of very short trajectories. So, while we collected a lot of trajectories for training this model, this is very uncommon for LPTM experiments (considering also the cost of the experiments). For this reason, we chose to work with fixed-length trajectories of 200 points (which can correspond to a 200-second video at 1 frame per second).

From a modeling perspective, the use of statistical methods in the loss function benefits from longer trajectories. Shorter trajectories do not provide enough data to accurately calculate key statistical features such as moments of displacement distributions or velocity autocorrelations, which are crucial for defining and characterizing stochastic processes. By using trajectories of 200 frames, we ensure robust statistical

representation while maintaining practical experimental and computational feasibility. We have included a concise justification of the choice of trajectory length in Section 2.1 of the revised manuscript:

“The trajectory length of 200 frames is a balanced choice to capture sufficient particle dynamics and accommodate datasets of varying sizes and collected across different video frame rates.”

5. Why did the authors choose to split the data into 1D trajectories? The two spatial dimensions should be highly correlated and this splitting should be properly motivated.

We thank the reviewer for raising this important point about the choice of splitting the data into 1D trajectories. After careful consideration of this feedback, we revised our approach to directly incorporate 2D trajectories as input, represented as tensors of size (2 x 200). Since the original experimental data is inherently two-dimensional, analyzing the trajectories in this form allows us to preserve the correlations between the x and y components, which are otherwise lost when splitting the data into separate 1D trajectories.

Previously, splitting trajectories into 1D components effectively doubled the dataset size, as each trajectory contributed its x and y components separately. To ensure a comparable dataset size for 2D analysis, we conducted additional experiments to collect ~17,000 experimental trajectories to add to the training dataset, covering a range of electron beam dose rates and particle sizes.

This transition to 2D input also required adapting all terms in the loss function to account for the position vector $r = (x, y)$, instead of just x , ensuring that the physics-informed loss accurately captures the 2D nature of the data.

We appreciate the reviewer’s suggestion, which prompted this significant improvement in our methodology.

6. It is not clear what the authors mean with “The loss function also incorporates an additional term that includes the mean value of the displacement calculated over a batch of trajectories in each training step, which ensures that large displacements are not taken at the same time points in every trajectory”.

We thank the reviewer for bringing up this point. The term in question was designed to calculate the error between the mean (across a training batch of trajectories) of displacements at each time point. Specifically, for each training step, the displacement vector was computed as $\Delta x(t) = x(t) - x(t - 1)$ for each trajectory, and the mean displacement over the batch was calculated for all 199 time points (corresponding to the 199 jumps in the trajectories of length 200). The error was then calculated between the mean jumps of the input trajectories and the mean jumps of the generated trajectories at each time point. This resulted in a 199-dimensional vector of errors, which were averaged to obtain the loss contribution from this term.

The intent of this term was to ensure that large displacements did not occur consistently at the same time points across trajectories in a batch, thereby promoting diversity in the temporal displacement patterns learned by the model. However, using the model performance metrics we defined based on the Reviewer’s first comment, we observed that this specific loss term did not improve the model performance. Consequently, we decided to remove this term from the loss function in the revised version of our manuscript, as it did not contribute to meaningful improvements in the model’s ability to capture the dynamics of the experimental data.

7. I find panels 2b and 2c unclear and rather difficult to read. Also, their discussion in the text (line 209) is not fully satisfactory.

We thank the reviewer for their feedback regarding panels 2b and 2c and the associated discussion in the text. In the revised manuscript, we have removed Figure 2 and its analysis, as mentioned in response to the Reviewer's first comment.

8. Figure 3 reports very interesting results, contributing to the interpretability of the method. However, the discussion is overly stretched and not fully supported by the data. Figure 3b shows a flat profile, except for the values of 2 and 3. Similarly, Fig. 3e, shows little variation for positive values of the latent variable. It is then difficult to confidently claim that these latent variables adequately capture non-Gaussianity.

We agree with the reviewer that the figure was confusing. In the revised manuscript, we have clarified the purpose and interpretation of this figure, incorporating changes to enhance its clarity and support for the findings.

To address the reviewer's concerns, we have updated Figure 3 to include the coefficient of determination (R^2 values) and lines of best fit for each scatter plot. These additions provide quantitative measures of the relationships between the latent variables and the statistical properties of trajectories. Specifically, Figure 3a now includes a matrix of R^2 values for second-degree polynomial fits between each latent variable and key statistical features, such as non-Gaussianity and velocity autocorrelation. This matrix offers a clearer and more comprehensive summary of the extent to which each latent variable captures these features.

The primary purpose of Figure 3 (shown below) is to demonstrate that specific latent variables learn certain statistical features while others do not. For instance, latent variable z_4 exhibits a strong relationship with velocity autocorrelation as shown in Figure 3g, but it does not correlate with non-Gaussianity, as evidenced by the flat profile in Figure 3f. Conversely, latent variables z_1 and z_5 demonstrate quadratic relationships with non-Gaussianity in the y and x components of the displacement distribution, respectively (Figures 3c and 3i), but do not capture temporal correlations (Figures 3d and 3j). These results emphasize that the latent variables are specialized and learn specific features of the trajectories rather than all possible features.

We thank the reviewer for their comment, which prompted us to refine the explanation of this section and make the purpose of Figure 3 more explicit. The updated figure and accompanying analysis now more clearly convey how different latent variables are associated with distinct statistical properties, highlighting LEONARDO's ability to learn meaningful and interpretable features from experimental data.

Fig. 3 Analysis of the physics learned by LEONARDO's latent variables. **a**, Matrix showing the coefficient of determination (R^2) for a second-degree polynomial fit between each latent variable z_i ($i = 1, \dots, 12$) and key statistical features of the trajectories: non-Gaussianity ($\xi(\tau = 1)$) of the x and y components (rows 1 and 2, respectively) and velocity autocorrelation ($C_v(\tau = 1)$) of the x and y components (rows 3 and 4, respectively). **b**, Probability density functions (PDF) of displacement distributions of trajectories generated by LEONARDO by modulating the sampled value of latent variable z_1 from -3 to $+3$, corresponding to ± 3 standard deviations away from the mean of the latent variable distribution. The PDFs of the x and y components are overlaid with Gaussian fits (dashed gray line for x and solid gray line for y). **c**, Scatter plot of non-Gaussianity ($\xi(\tau = 1)$) versus latent variable z_1 . The black dashed and solid lines represent second-degree polynomial fits to the x and y components, respectively. **d**, Scatter plot of velocity autocorrelation ($C_v(\tau = 1)$) versus latent variable z_1 , with second-degree polynomial fits as in (c). **e-g**, Same analysis as (b)-(d), but for latent variable z_4 . **h-j**, Same analysis as (b)-(d), but for latent variable z_5 .

9. The statement at the end of section 2.2 “Thus, z_2 essentially captures the heterogeneity of the energy landscape of the environment that the particles interact with as they move, while z_{12} captures the local viscoelasticity of the environment surrounding the particles. This observation confirms that the latent variables of Leonardo learned specific physics about the nanoparticles and their interactions with their surrounding environment due to the implementation of the physics-informed loss function, enabling it to distinguish between processes of different mechanisms, including the complex motion of nanoparticles in LPTM.” is quite speculative and should be stated as such. What the authors find are intriguing, suggestive indications and correlations between latent variables and some statistical features. These are definitely not hard evidence. The connection between the statistical features and the physical picture is very speculative. Therefore the authors’ claims should be less confident and clearly stated to be, though perhaps plausible, speculations.

We thank the reviewer for their thoughtful comment and agree that the wording in the original manuscript was overly confident. In response, we have revised the section to better reflect the nature of our findings. Specifically, we now emphasize that the relationships observed between latent variables and statistical features are suggestive rather than definitive. The revised text reads:

"Thus, z_1 and z_5 primarily learn the non-Gaussianity at $\tau = 1$, which is suggestive of the heterogeneity of the energy landscape that the particles traverse. Similarly, z_4 captures the velocity autocorrelation at $\tau = 1$, which is indicative of the viscoelasticity of the environment surrounding the particles. These results demonstrate how LEONARDO's latent variables capture statistical properties that relate to the physical processes governing particle motion in LPTM. This capability allows LEONARDO to probe how experimental factors, such as particle size and electron beam dose rate, influence the interaction energy landscape in the liquid cell, as explored in the next section."

10. Figure 4, although interesting, is mostly a visual, non-quantitative one and its discussion is handwaving. Its scientific significance is, therefore limited.

We thank the reviewer for their feedback on Figure 4. We agree with the reviewer that the figure, while visually informative, lacked sufficient quantitative analysis to provide substantial scientific significance. In light of this, we have removed Figure 4 and its associated discussion from the manuscript.

To address the visualization and exploration of generated trajectories in a more interactive and flexible manner, we have released a Jupyter Notebook as part of the code repository on github. This notebook allows users to generate and visualize trajectories using LEONARDO's latent space. This approach provides greater flexibility for users to explore the model's capabilities and observe trajectory behavior under various conditions, fulfilling the intent of the original figure in a more interactive and user-driven way.

11. In the beginning of section 2.4 the authors claim that two latent variables capture the non-Gaussianity and the negative autocorrelation of the trajectories. To convincingly show this the authors should provide more quantitative evidence than the one presented in Fig. 3. Scatter plots similar to the ones of Fig. 2b and correlation coefficients may serve this purpose well.

We thank the reviewer for their insightful comment, which has prompted us to refine and improve the analysis in the revised manuscript. In response to the suggestion for more quantitative evidence to support the claim that certain latent variables capture non-Gaussianity and velocity autocorrelation, we have made modifications to the analysis in Section 2.3 and updated Figure 3 accordingly.

The specific changes made to Figure 3 and its analysis, including the addition of R^2 values and lines of best fit, have been described in detail in our response to Reviewer Comment 8. These changes directly address the reviewer's request for more rigorous and quantitative evidence, enhancing the clarity and interpretability of our findings. We thank the Reviewer for their valuable suggestion, which has greatly improved this section.

12. How would the plots in Fig. 5 look if non-Gaussianity and the magnitude of the negative peak of autocorrelation were plotted? This would help understand to what extent the correlation between the two latent variables and these statistical features, reported in Fig. 3, is present in experimental data. This would

also help the reader appreciate the relation and differences between using Leonardo and the usual statistical measures.

We thank the reviewer for their insightful suggestion to include plots comparing non-Gaussianity and the magnitude of negative velocity autocorrelation with the latent variables, as this helps assess the correlations identified in Figure 3 and better connects LEONARDO's results to statistical measures.

In response, we revised the previous Figure 5 (now Figure 4) and its associated analysis in Section 2.4 to address this point. The updated figure now includes direct comparisons of non-Gaussianity and velocity autocorrelation distributions with the corresponding latent variable distributions across different experimental conditions (particle size and electron beam dose rate). These comparisons help illustrate the extent to which LEONARDO's latent variables encode these statistical properties in unseen experimental data.

We thank the reviewer for their suggestion, which has significantly improved the clarity and scientific value of this section.

Fig. 4 Characterization of the interaction energy landscape in LPTEM using LEONARDO. **a**, Probability densities of the values of $\max(z_1^2, z_5^2)$ for experimental trajectories collected at electron beam dose rates of 20 and 30 $e^-/\text{\AA}^2\cdot\text{s}$ and particle size of 60 nm. **b**, Probability densities of the values of $\max(z_1^2, z_5^2)$ for experimental trajectories collected at particle sizes of 40 nm and 60 nm at an electron beam dose rate of 35 $e^-/\text{\AA}^2\cdot\text{s}$. **c**, Probability densities of the values of z_4 for the same experimental trajectories as in (a). **d**, Probability densities of the values of z_4 for the same experimental trajectories as in (b). **e**, Probability densities of the values of non-Gaussianity at $\tau = 1$ for the same experimental trajectories as in (a) and (c). **f**, Probability densities of the values of non-Gaussianity at $\tau = 1$ for the same experimental trajectories as in (b) and (d). **g**, Probability densities of the values of velocity autocorrelation at $\tau = 1$ for the same experimental trajectories as in (a), (c), and (e). **h**, Probability densities of the values of velocity autocorrelation at $\tau = 1$ for the same experimental trajectories as in (b), (d), and (f).

13. The discussion about the complex potential landscape in which the particles may be moving is too qualitative. Especially, the evidence about the hierarchy of potential wells is very weak. To make it stronger, the authors should perform an analysis similar to that shown in Fig. 5 with simulations of FBM and CTRW. They should also show the correlation between waiting times and z_{12} . Otherwise, the authors should carefully rephrase the paragraph to make it clear that they are just suggesting the landscape and not demonstrating. Similarly, the last sentence in the conclusion should be removed. To make it really

quantitative, the authors might consider explicitly simulating the hierarchy of potential they discuss, although, this may be beyond the scope of this paper.

We thank the reviewer for their valuable feedback and for prompting us to critically re-evaluate the discussion of the hierarchical energy landscape in our original manuscript. Upon reflection, we agree that the evidence for a hierarchy of potential wells was not sufficiently quantitative and required further justification. As a result, we have removed the discussion of hierarchical energy landscapes entirely from the revised manuscript.

In its place, we now provide a more quantitative and focused analysis in Section 2.4, as shown in the new Figure 4. This revised analysis examines how latent variables z_1, z_4 , and z_5 – which capture statistical features such as non-Gaussianity and velocity autocorrelation—respond to changes in experimental parameters, namely electron beam dose rate and particle size. By directly comparing the distributions of these latent variables with the corresponding statistical properties, such as non-Gaussianity and velocity autocorrelation at $\tau = 1$, we present a more quantitative characterization of the interaction energy landscape. This approach strengthens the scientific clarity and avoids speculative interpretations.

We thank the reviewer for helping us improve the manuscript by encouraging a more focused and quantitative discussion.

Minor comments

- In line 76, I find the choice of the pronoun “they” when referring to a generative AI odd.

We thank the Reviewer for pointing out this error. This has been corrected in the revised manuscript.

- In line 125, it is unclear what the authors mean by “within the limit of normal diffusion”.

We thank the reviewer for pointing out the unclear phrasing. This statement was part of the discussion related to simulated hybrid trajectories, which have since been removed from the revised manuscript.

- In line 145, physics-informed machine learning does not only adjust the hyperparameters but affects the whole training.

We thank the reviewer for pointing out the need for clarification regarding the role of physics-informed machine learning. In the revised manuscript, we have re-worded the sentence to more accurately reflect the nature of physics-informed machine learning:

"Physics-informed machine learning has recently gained traction in applications where the partially known physics of the input data can be incorporated into the model's loss function. This approach constrains the learning process with physical laws, ensuring the model captures the underlying physics of the system."

- The definition and introduction of the statistical features around line 149 could be streamlined, for example, by defining μ and σ from the start.

We thank the reviewer for suggesting this edit to the manuscript. In the revised manuscript, we have ensured that no terms are defined multiple times, or defined before they are being used in a mathematical relationship.

- In line 201, the fact that FBM and CTRW are very different mechanisms makes it easier to tell them apart, not harder, as the authors seem to suggest.

We thank the reviewer for this observation. This statement was part of the analysis and discussion related to the previous Figure 2. As noted in our responses to the Reviewer's first comment, we have removed Figure 2 and its associated analysis in the revised manuscript.

- It would be helpful to have some more details about the p-values given in lines 215, 216. What is the null hypothesis there? I think the whole discussion of Fig. 2 could be streamlined. It would also be useful to recall a definition of the absolute correlation coefficient.

We thank the reviewer for this comment. The discussion referenced was part of the analysis associated with the previous Figure 2, which has since been removed in the revised manuscript. However, we have taken the reviewer's suggestion to streamline the analysis and provide clarity. For the revised Figure 3, which now focuses on experimental data, we have included the coefficient of determination R^2 to quantify the relationships between latent variables and statistical features. For that, we have provided the definition and formula for calculating R^2 in the Methods section to ensure transparency and reproducibility.

- The authors claim "Beyond merely simulating known behaviors, the model has other potential applications, including the exploration of novel nanoparticle dynamics, assisting in hypothesis generation for experimental settings, and serving as a predictive tool for interpreting complex diffusive behaviors." but do not justify the claim nor explain how they would do what they suggest.

We thank the reviewer for highlighting this point. Upon reflection, we agree that the original claim about hypothesis generation was not sufficiently justified in the manuscript. This was part of our broader vision for the potential applications of generative AI models when trained on large and diverse datasets of experimental trajectories. However, we recognize that this capability is aspirational and would require extensive further development, including the collection of a significantly larger experimental dataset, with more experimental variables, and refinements to the model itself. As such, we have revised the manuscript to remove this point, and instead focus solely on the demonstrated potential use cases of LEONARDO.

- Typo in line 324, it should be Fig. 5 d, i.

We thank the Reviewer for pointing out this error. We made sure to reference the correct figures in the revised Section 2.4.

- Reference 18 contains a typo.

We thank the Reviewer for the comment, we have fixed this typo.

We sincerely thank Reviewer 2 for their constructive feedback, which has significantly improved the scientific clarity and rigor of our manuscript. Below, we summarize our responses to the key points raised:

1. **Performance metrics for model evaluation:**

We have introduced two quantitative performance metrics to assess LEONARDO's ability to generate trajectories resembling experimental data. The first metric uses the Fréchet distance (FD) computed in a learned feature space derived from MoNet, a classifier trained to distinguish between diffusion classes. The second metric evaluates the percentage of LEONARDO-generated trajectories classified as LPTM trajectories by MoNet, with over 95% being correctly classified. These metrics, detailed in Section 2.2 of the revised manuscript, provide robust evidence of LEONARDO's performance.

2. **Choice of statistical features:**

In response to the suggestion to assess additional statistical features, we conducted a comprehensive ablation study inspired by the CONDOR framework. This study evaluated features such as wavelet coefficients, Fourier transforms, and the positional autocorrelation function. Our results showed that incorporating the positional autocorrelation function and the median displacement term into the loss function further improved model performance. The remaining statistical features did not significantly enhance the model's ability to generate accurate trajectories.

3. **Learning from short trajectories and limitations:**

We clarified that LEONARDO analyzes longer trajectories by segmenting them into shorter, fixed-length pieces while retaining the ability to study longer time scales through subsampling. This approach balances computational feasibility with the model's capacity to capture both short- and long-range trajectory behavior. We revised the text in Section 2.1 to better explain this point.

4. **Trajectory length (200 frames):**

The choice of 200 frames was guided by experimental and modeling considerations. Experimentally, 200-frame trajectories are achievable in microscopy without introducing tracking artifacts or electron beam damage. From a modeling perspective, 200 frames are sufficient to compute robust statistical properties while ensuring computational efficiency. This justification has been added to Section 2.1.

5. **Use of 2D trajectories:**

In response to concerns about splitting trajectories into 1D components, we revised our approach to use 2D trajectories as input. This preserves correlations between the x and y components of the trajectories. To achieve a comparable dataset size, we collected additional experimental trajectories and updated the loss function terms to account for the 2D positional vector.

6. **Removal of additional loss term:**

The additional loss term, which computed the mean displacement error across batches, was removed after evaluating its contribution to model performance. Our analysis showed that this term did not improve performance metrics, and it was therefore excluded from the final model.

7. **Figure 2 and associated analysis:**

We removed Figure 2 and its associated analysis from the manuscript based on feedback from both reviewers. This change simplifies the manuscript and ensures that the analysis focuses exclusively on experimental data.

8. **Figure 3 and quantitative evidence:**

To provide more quantitative evidence for the relationships between latent variables and statistical features, we updated Figure 3 to include lines of best fit and the coefficient of determination (R^2). This addition clarifies the extent to which latent variables capture statistical properties such as non-Gaussianity and velocity autocorrelation. These improvements are detailed in Section 2.3.

9. **Speculative claims regarding latent variables:**

We revised the wording in Section 2.3 to ensure that our claims are appropriately cautious and clearly presented as suggestive rather than definitive. The revised text now highlights that latent variables capture statistical properties that relate to the physical processes governing particle motion in LPTM.

10. **Figure 4 and improved quantitative analysis:**

We replaced the previous Figure 4 with a more quantitative analysis in the new Figure 4. This figure compares the distributions of latent variables with the corresponding statistical properties (non-Gaussianity and velocity autocorrelation) across different experimental conditions. This revised analysis avoids speculative claims and provides clear, quantitative evidence.

11. **Clarification of statistical feature relationships:**

In response to the suggestion to compare statistical measures directly with latent variables, we revised Section 2.4 to include comparisons of non-Gaussianity and velocity autocorrelation distributions with the latent variable distributions. This strengthens the link between LEONARDO's learned latent space and the statistical properties of trajectories.

12. **Hierarchy of potential wells:**

We removed the discussion of hierarchical energy landscapes, as we agreed with the reviewer that the evidence for this claim was insufficient. Instead, Section 2.4 now focuses on a more quantitative analysis of the latent variables' relationships with particle size and electron beam dose rate.

References

- [1] G. Fernández-Fernández, C. Manzo, M. Lewenstein, A. Dauphin and G. Muñoz-Gil, "Learning minimal representations of stochastic processes with variational autoencoders," *PHYSICAL REVIEW E*, 2024.
- [2] M. Heusel, H. Ramsauer, T. Unterthiner and B. Nessler, "GANs Trained by a Two Time-Scale Update Rule Converge to a Local Nash Equilibrium," *Arxiv*, 2017.
- [3] V. Jamali, C. Hargus, A. Ben-Moshe and P. A. Alivisatos, "Anomalous nanoparticle surface diffusion in LCTEM is revealed by deep learning-assisted analysis," *PNAS*, 2021.

[4] G. Muñoz-Gil, G. Volpe, M. . A. Garcia-March and C. Manzo, "Objective comparison of methods to decode anomalous diffusion," *Nature Communications*, 2021.

[5] A. Gentili and G. Volpe, "Characterization of anomalous diffusion classical statistics powered by deep learning (CONDOR)," *Journal of Physics A: Mathematical and Theoretical*, 2021.

Appendix

This section provides the figures illustrating how the losses for all the statistical features listed in the response to Reviewer 2 Comment 2 evolved over intermediate baseline models, which were trained for 1, 20, 40, 80, 120, and 160 epochs and the final baseline model, trained for 200 epochs.

1. Median of displacements of trajectories

2. Mean, median, variance, skewness, and kurtosis of the absolute values of the normalized displacements.

3. Mean, median, variance, skewness, and kurtosis of the eight vectors of the absolute values of displacements taken at eight different time lags.

4. Median of the ratios between consecutive normalized displacements.

5. Mean, median, variance, skewness, and kurtosis of the absolute values of the discrete Fourier Transform of the displacements centered at the zero-frequency component and normalized to the trajectory length.

6. Mean, median, variance, skewness, and kurtosis of the integrated power spectral density (IPSD) as described in [5].

7. Mean, median, variance, skewness, and kurtosis of the absolute values of the variations of the time-averaged mean squared displacement normalized to the corresponding time lag.

8. Mean, median, variance, skewness, and kurtosis of the absolute values of wavelet coefficients using the Morlet wavelet function.

9. The integral IACF as described in [5].

10. The level of frequency asymmetry as described in [5].

11. The level of time asymmetry as described in [5].

12. Absence of abrupt changes as described in [5].

13. Positional autocorrelation function

REVIEWER 1

I highly appreciate the effort that the authors have put in answering my concern. I think the manuscript has greatly improved. I still have a few questions, mostly related to the core ideas behind the work, and what we can actually learn from the trained machine.

Response to Comment 1.1

I thank the authors for their response. I agree with their perspective on the Fernández-Fernández et al. loss, only being valid for Gaussian processes. My comment on the probabilistic loss was indeed more general: given that we are compressing a stochastic signal in the latent space (3 neurons in this case), it is impossible, from an informational perspective, to recover the original input signal, as this was random. It is possible however, to generate trajectories that come from the same distribution. This is what was shown in Fernández-Fernández et al., and what motivates the adequacy of the probabilistic loss vs. an MSE loss, the latter making the training of the network ill-defined by the previous argument.

I believe the rest of the terms in LEONARDO's loss are taking this into account, and the MSE is indeed not needed to reach the results shown in the paper. May this be highlighted by the fact that w_1 in S6 (Supplementary Material) is at least order of magnitude smaller than the rest of the weights?

We thank the reviewer for the insightful comments and fully agree that, given the inherently stochastic nature of the input trajectories and the limited capacity of the latent space, it is more meaningful to reproduce the statistical properties of the data rather than aiming for exact reconstruction. This perspective is indeed stated in the opening paragraph of the “Physics-informed loss function” subsection in Section 2 of our manuscript.

To further clarify this point, we have explicitly noted in the revised Main text:

“Thus, we designed a physics-informed loss function that includes terms customized to learn stochastic trajectories in addition to the standard loss terms in a VAE and we assigned a lower weight to the standard MSE-based reconstruction term to minimize its contribution to the total loss (see Supplementary Information for details of each loss term)”

Additionally, in the revised Supplementary Information, we added:

“The very low weight assigned to w_1 (corresponding to the MSE-based reconstruction loss) emphasizes that the contribution of the MSE loss term is small compared to the physics-informed loss terms. This choice of weight reflects our focus on reproducing the statistical distribution of the trajectories rather than achieving an exact point-wise reconstruction.”

Response to Comment 1.2

I thank the authors for taking my comments into account about the model's architecture. On this, just a small comment: in Fig. 1b it is not clear that there is a convolutional layer after the attention block of the decoder, as it seems that that attention block directly outputs the trajectory.

We thank the reviewer for the comment regarding the decoder architecture. For further clarity, we have updated Figure 1b (attached below) to show a more detailed architecture, including the convolutional layer following the attention block, and also added more details in Methods Section 4.3 on line 440 of the Main text to further emphasize that the final convolutional layer plays a key role in preserving temporal correlations in the generated trajectories:

“..., which serves to preserve temporal correlations in the trajectories.”

About the Fréchet distance calculated, what is the typical value for matching distributions? It may be useful for the reader to know what is the FD at the diagonal (e.g. FBM vs FBM, CTRW vs CTRW,...) to then be able to better compare what the 0.547 of the Generated vs. LPTM means.

We thank the reviewer for the constructive suggestion regarding the interpretation of the Fréchet distance (FD) values. We agree with the reviewer that providing a comparison to intra-class FD scores is important to help readers better understand and contextualize the significance of the FD score between LEONARDO-generated and experimental LPTM trajectories.

We have now added a new figure in the Supplementary Information shown below (Figure S4), which shows a lower triangular matrix of FD values computed between two independent batches of each diffusion class. This provides a clear reference for the typical intra-class FD scores, which serve as a baseline for interpreting the FD score between LEONARDO and LPTM.

In addition, we have added the following clarifying sentence to the Main text:

“The scores computed between two independent batches of the same diffusion class (e.g., FBM vs. FBM, CTRW vs. CTRW, etc.) range from 0.19 to 1.74 (see Figure S4), providing a baseline for interpreting the Fréchet distance values.”

We also want to note that to ensure consistency in feature extraction, we applied the same normalization procedure for both training MoNet2.0 and evaluating trajectories in the FD calculations in the revised manuscript. The updated FD matrix shown below has been incorporated into the Main text, and the revised values continue to support the conclusion that LEONARDO-generated trajectories are closest to experimental LPTEM data.

Fig. S4: Fréchet Distance lower triangular matrix showing scores between pairs of diffusion class including intra-class scores. The second-last layer output of MoNet2.0 is used to compute the FD scores between different diffusion classes and between batches of the same class (intra-class scores). The matrix shows that the FD scores between LPTEM and LEONARDO-generated trajectories are significantly lower than the scores between other diffusion classes, while intra-class scores, ranging from 0.26 to 1.12, provide a lower bound for contextual comparison.

Response to Comment 1.3 and 1.4

I sincerely thank the authors for considering my comments and the extensive work they put into answering my questions. I have still few concerns relating this two comments:

- 1) Related to the factors of variation learned by the VAE: the new plots in Fig.4 a+e, b+f,... highlight that there is a connection between the latent neurons and the dose rates and nanorods sizes. Reformulating my comment from the previous review round: what happens when you plot the latent neurons in Fig3c-j w.r.t. to the two experimental parameters? Ultimately, has the model learned to represent in the latent space: a) the size and dose or b) the non-Gaussianity and the velocity autocorrelation? I understand that these two sets are proportional to each other (as shown in Fig. 4), but it would be interesting to know if the authors have any insight on whether the VAE has learned one or the other.
- 2) Minor comment: both in Fig. 3 and 4, the authors show the value z of the latent neuron. However, it is usually sufficient to look at the value μ , as the value of σ is typically independent of the input sample and gives no information about it. It is indeed μ which is encoding the factors of variations. By plotting it instead of z , you usually eliminate unnecessary fluctuations in the plots and can more easily show what is exactly learned. For instance, in Fig4d, the purple plot arises from having multiple μ distributed across the domain or because of the variance σ^2 ? Similar for the rest of the plots.
- 3) Why do three neurons survive, rather than two? Is the diffusion inhomogeneous? If not, why is the non-gaussianity for each component separated in the latent space? If the diffusion is homogeneous, I would have expected only two neurons. If it is not, where is this inhomogeneity coming from? I understand that in this case it would be independent of the dose and size, and would help clarify my point 1) above.

We sincerely thank the reviewer for the thoughtful follow-up questions and for the opportunity to further clarify these important aspects of our work.

- 1) We would like to emphasize that LEONARDO is designed to model LPTEM trajectories by learning their statistical properties, not experimental parameters directly. Specifically, the physics-informed loss function drives the latent space to capture statistical properties such as non-Gaussianity and velocity autocorrelation, which reflect heterogeneity and viscoelasticity of the environment – the two environmental properties we aimed to investigate in this work. These environmental properties are, in turn, influenced by the dose rate and particle size, as shown in Figure 4. In fact, in Figure 4 we show that there is indeed a connection between latent variables and statistical properties, as well as these experimental parameters as expected (the experimental data comes from solutions of nanorods of 40 and 60 nm in size and as shown in Figure 4, the UMAP plots of latent variables show that the latent variables are capable of clustering them into two clusters (see figure attached below)), and we discussed that in detail in Section 2.4 of the revised manuscript. To further illustrate this and for complementary visualization of the histogram plots in Figure 4, we have included in the Supplementary Information a set of pairwise scatter plots between μ_1 , μ_4 , and μ_5 under the two sets of dose and size conditions (Supplementary Figure S6), which is also attached below.
- 2) We agree that plotting the latent mean μ rather than the sampled latent variable can reduce unnecessary fluctuations arising from the sampling process. Accordingly, we have regenerated the plots in Figure 4 (attached below) using μ values and updated the manuscript. However, we note that in Figure 3, we need to explicitly vary the value of a latent variable to examine its effect on the generated trajectories, so the concept of plotting the encoded μ does not apply in that case.
- 3) Multiple latent variables survive in LEONARDO, capturing different statistical properties to serve as the most effective generative model for stochastic trajectories from LPTEM. In this work, we

focused on three of the latent variables— z_1 , z_4 , and z_5 — and show that they encode the properties of interest related to the interaction energy landscape of LPTEM, that are non-Gaussianity and velocity autocorrelation. Non-Gaussianity in the x and y components vary independently in the experimental dataset as expected—note that from the experimental point of view there is no reason for a particle to take a symmetric long jump in x and y simultaneously (long jumps lead to non-Gaussianity). In some cases the x -component of a large jump is much larger than the y -component, and vice versa, leading the model to represent them separately in z_1 (for y) and z_5 (for x). In contrast, velocity autocorrelation does not vary independently between x and y in our dataset, so it is captured by a single latent variable, z_4 . This is because a particle taking anti-correlated step does the same thing for both x and y coordinates simultaneously due to the same level of “confinement” in x and y . While other latent variables also encode meaningful statistical properties (as discussed in the response to reviewer comment 1.5 below), these three are directly tied to heterogeneity and viscoelasticity of the LPTEM environment which are the focus of this study.

Fig. 4 Characterization of the interaction energy landscape in LPTEM using LEONARDO. **a**, Probability densities of the values of $\langle \mu_1, \mu_5 \rangle$ for experimental trajectories collected at electron beam dose rates of 20 and 35 $e^-/\text{\AA}^2\text{-s}$ for particle size of 60 nm. **b**, Probability densities of the values of μ_4 for the same experimental trajectories as in (a). **c**, Probability densities of the values of $\langle \mu_1, \mu_5 \rangle$ for experimental trajectories collected at particle sizes of 40 nm and 60 nm at an electron beam dose rate of 35 $e^-/\text{\AA}^2\text{-s}$. **d**, Probability densities of the values of μ_4 for the same experimental trajectories as in (c). **e**, Probability densities of the values of non-Gaussianity (averaged over the x and y components) at $\tau = 1$ for the same experimental trajectories as in (a). **f**, Probability densities of the values of velocity autocorrelation (averaged over the x and y components) at $\tau = 1$ for the same experimental trajectories as in (b). **g**, Probability densities of the values of non-Gaussianity at $\tau = 1$ for the same experimental trajectories as in (c). **h**, Probability densities of the values of velocity autocorrelation at $\tau = 1$ for the same experimental trajectories as in (d). **i**, UMAP embeddings of all twelve latent variables for the same experimental trajectories as in (a) and (c), color-coded by dose rate. **j**, UMAP embeddings of all twelve latent variables for the same experimental trajectories as in (a), (c), and (i), color-coded by non-Gaussianity ($\tau = 1$) on a symmetric logarithmic scale (SymLogNorm). **k**, UMAP embeddings of all twelve latent variables for the same experimental trajectories as in (a), (c), and (i), color-coded by velocity autocorrelation ($\tau = 1$). **l**, UMAP embeddings of all twelve latent variables for the same experimental trajectories as in (c) and (g), color-coded by particle size. **m**, UMAP embeddings of all twelve latent variables for the same experimental trajectories as in (c), (g), and (l), color-coded by non-Gaussianity ($\tau = 1$) on a symmetric logarithmic scale (SymLogNorm). **n**, UMAP embeddings of all twelve latent variables for the same experimental trajectories as in (c), (g), and (l), color-coded by velocity autocorrelation ($\tau = 1$).

Fig. S6: LEONARDO latent space scatter plots. a, Scatter plot of the values of μ_4 versus μ_1 for experimental trajectories collected at electron beam dose rates of 20 and 35 $e^-/\text{\AA}^2\cdot\text{s}$ for a fixed particle size of 60 nm. b, Scatter plot of the values of μ_4 versus μ_5 for the same experimental trajectories as in (a). c, Scatter plot of the values of μ_1 versus μ_5 for the same experimental trajectories as in (a) and (b). d, Scatter plot of the values of μ_4 versus μ_1 for experimental trajectories collected at particle sizes of 40 nm and 60 nm at a fixed electron beam dose rate of 35 $e^-/\text{\AA}^2\cdot\text{s}$. e, Scatter plot of the values of μ_4 versus μ_5 for the same experimental trajectories as in (d). f, Scatter plot of the values of μ_1 versus μ_5 for the same experimental trajectories as in (d) and (e).

Response to Comment 1.5

First, as commented above, the variance is typically almost constant for any input sample. As soon as the VAE “decides” that a latent neuron encodes a factor of variation, it just lowers its value to something close to zero for any input sample.

Given that, I am very surprised by the values of the variances presented by the authors. Typically, the values of the variances of neurons that encode valid information (e.g. z_1 , z_4 and z_5) are orders of magnitude smaller. It is usual to have $\log(\sigma^2) \ll -4$. For instance, let’s consider that LEONARDO get’s a trajectory with a given $C_v = -0.2$. Then, the encoders processes this information and sets $\mu_4 \rightarrow 0$ (from Fig. 3g). Now, because of the big variance, when you sample z_4 the information of the input C_v is completely noised out, as with that variance it’s almost impossible to distinguish e.g. $\mu_4 = 0$ and $\mu_4 = 0,5$. This error will be propagated to the decoder, which will then lead to a bad reconstruction loss.

This is even more surprising for neurons z_1 and z_5 , which both have a much bigger variance. This means that no matter what is encoded in that neuron, it is almost completely noised out by such big variances.

Are the numbers presented correct? If so, can the authors comment on this? I suspect that the reconstruction loss should be very high because of this.

We thank the reviewer for the valuable feedback regarding the latent variable variances. We must first acknowledge that we initially misunderstood the comment in the first revision. When the reviewer mentioned that “the variance of most of the neurons” should be close to 1, we interpreted this as referring to the variance of the latent z values themselves – essentially, how spread out the data is on both sides of the regression line in Figure 3g (i.e., the residual variance) – rather than the variance provided directly by the encoder. We now understand that the reviewer was referring to the variance as given by the encoder (i.e., taking the exponential of $\log \sigma^2$), which is a more direct indicator of whether a latent neuron is carrying meaningful information.

In the revised version, we now used the correct methodology by encoding the full test set, as described below:

1. Encode the full test set: Pass all test trajectories through the model to obtain the $z, \mu, \log \sigma^2$, and output trajectories.
2. Convert the $\log \sigma^2$ values to variance by computing $e^{\log \sigma^2}$ for each latent neuron.
3. Average these log-variance and variance values over all trajectories for each latent variable.

This updated approach gives the following results:

Latent variable	z_1	z_2	z_3	z_4	z_5	z_6	z_7	z_8	z_9	z_{10}	z_{11}	z_{12}
$\log \sigma^2$	-4.57	-0.06	-1.53	-2.76	-4.79	-2.27	-0.04	-2.21	-0.04	-2.76	-0.03	-0.06
Variance	0.011	0.95	0.26	0.067	0.0088	0.11	0.96	0.12	0.96	0.065	0.97	0.94

These results confirm that latent variables z_1, z_4 , and z_5 which are related to non-Gaussianity and velocity autocorrelation as shown in Figure 3a, have low variances. We also note that there are other latent variables with low variances, which is expected given the design of our physics-informed loss function and their role in encoding statistical properties important for capturing the full diversity of the physical processes in LPTEM. For example, in case of z_{10} , it captures the correlation between x and y components of the trajectories, which is important for accurately modeling LPTEM trajectories but falls outside the scope of the properties related to interaction energy landscape analyzed in this study.

To provide more clarity about these variances in our work we added the table above as a Supplementary Table. We also added the following sentences at the end of Section 2.3 in the revised manuscript:

“While we focused on three latent variables due to their relevance to the heterogeneity of the energy landscape and the viscoelasticity of LPTEM environment, other latent variables also encode statistical properties important for capturing the full diversity of the physical processes in LPTEM. The complete table of the variances for all latent variables, computed from the encoder outputs on the test set, is provided in Table S1 of the Supplementary Information.”

We thank the reviewer for contributing to this work in a positive way through these valuable suggestions.

REVIEWER 2

The authors have performed extensive work to address the reviewers' comments. I find this second version of the manuscript has markedly improved. However, some of the changes included require additional explanation before I can recommend the manuscript for publication. Below, I will detail the points that need to be discussed.

MAIN COMMENTS

Quality assessment of the generated trajectories

The authors developed two ingenious ways of quantifying how accurately the generated trajectories represent LPTM trajectories. However, their assessments rely on another neural network, which introduces an additional black box making the procedure difficult to scrutinise. Could the author also report simply the “distance” between the generated LPTM trajectories and the original ones in terms of the statistical features they use for their loss function? For example, the Mean squared error of the mean, of the variance of the displacements, of the skewness, etc.

To give a reference, one could compare them with the same metrics for other anomalous diffusion models, such as the five models analysed by the authors in Fig. 2.

We thank the reviewer for appreciating our work and for the thoughtful suggestions. We would like to note that directly comparing the generated and experimental LPTEM trajectories using the same statistical features that were part of the loss function would not provide a fully fair and independent evaluation metric. Since LEONARDO was explicitly optimized to minimize differences in those exact features as part of the physics-informed loss function, such a comparison would inevitably show a very low distance between the generated and LPTEM trajectories compared to the scores with other diffusion classes.

To ensure a more unbiased and broadly accepted evaluation, we instead applied the FD method, a well-established and widely used metric for evaluating generative models (commonly referred to as FID when applied to images using the Inception-v3 model, however here we used MoNet2.0 in our case since our data consists of particle trajectories rather than images). This method compares the distributions of real and generated trajectories in a learned feature space, rather than directly re-using the hand-crafted features from the loss function. This approach follows established best practices in the generative AI community, where FD is a standard method for assessing the quality of generative models.

Relation between the classifiers and the Fréchet Distance

I find it difficult to draw a precise connection between the conclusions suggested by the classifier and by the Fréchet Distance (FD). From the FD metrics, the LPTM trajectories seem rather close to the CTRW ones. However, the classifier never mistakes an LPTM trajectory for a CTRW one. It does instead mistake it for BM and FBM in about 1% of the cases each. Therefore, these two ways of determining how close trajectories are seem inconsistent.

We thank the reviewer for this insightful question. We appreciate the opportunity to clarify the distinction between the classifier confusion matrix and the Fréchet Distance (FD) analysis.

We would like to emphasize that the classifier confusion matrix and the FD between feature vectors are not expected to correlate directly, as they measure fundamentally different properties. The classifier confusion matrix reflects how the trained network separates classes based on the most discriminative features learned

during supervised training. In contrast, the FD quantifies the global statistical similarity between two distributions of trajectories by comparing the overall means and covariances of their feature vectors. This captures how similar the two distributions are when considering all measured properties of the trajectories, rather than just the subset of features that are most useful for classification. Therefore, the classifier’s decision boundary may clearly separate two distributions, even if their learned features are close (leading to a low FD). This is evident, for example, in the relatively low FD between Levy Walk and Brownian Motion, even though the classifier separates them with perfect accuracy.

We also want to note that to ensure consistency in feature extraction, we applied the same normalization procedure for both training MoNet2.0 and evaluating trajectories in the FD calculations in the revised manuscript. The updated FD matrix shown below has been incorporated into the Main text, and the revised values continue to support the conclusion that LEONARDO-generated trajectories are closest to experimental LPTEM data.

I am also confused because in their first submission, the authors modelled LPTM as a mixture of FBM and CTRW. This seems at odds with the fact that the classifier never mistakes an LPTM trajectory for a CTRW one or vice versa.

We thank the reviewer for this question. In our first submission, we modeled LPTEM trajectories as a combination of FBM and CTRW based on experimental observations—specifically, the presence of large jumps (suggestive of CTRW-like behavior) and anticorrelated motion (characteristic of FBM). However, based on the reviewer’s feedback in the previous revision, we moved away from this assumption and focused solely on training the model on the experimental data from LPTEM experiments. The updated analysis, including the classifier results, now confirms that this approach provides a more robust way to characterize LPTEM trajectories without imposing a predefined model and we thank the reviewers for this constructive suggestion.

UMAP

Could the authors present a UMAP representation of the latent variables colour-coded for the different experimental conditions as the ones shown in Fig. 4? Colour-coding in terms of the non-Gaussianity or autocorrelation could also be informative.

We thank the reviewer for this suggestion. We agree that UMAP visualizations of the latent variables color-coded by experimental conditions, non-Gaussianity, and velocity autocorrelation provide valuable additional insights. In response, we have added six UMAP plots: one for the dose-effect study, one for the size-effect study, and two each for non-Gaussianity and velocity autocorrelation statistics. The fully revised Figure 4 is presented below.

Fig. 4 Characterization of the interaction energy landscape in LPTEM using LEONARDO. a, Probability densities of the values of $\langle \mu_1, \mu_5 \rangle$ for experimental trajectories collected at electron beam dose rates of 20 and 35 $e^-/\text{\AA}^2\cdot\text{s}$ for particle size of 60 nm. b, Probability densities of the values of μ_4 for the same experimental trajectories as in (a). c, Probability densities of the values of $\langle \mu_1, \mu_5 \rangle$ for experimental trajectories collected at particle sizes of 40 nm and 60 nm at an electron beam dose rate of 35 $e^-/\text{\AA}^2\cdot\text{s}$. d, Probability densities of the values of μ_4 for the same experimental trajectories as in (c). e, Probability densities of the values of non-Gaussianity (averaged over the x and y components) at $\tau = 1$ for the same experimental trajectories as in (a). f, Probability densities of the values of velocity autocorrelation (averaged over the x and y components) at $\tau = 1$ for the same experimental trajectories as in (b). g, Probability densities of the values of non-Gaussianity at $\tau = 1$ for the same experimental trajectories as in (c). h, Probability densities of the values of velocity autocorrelation at $\tau = 1$ for the same experimental trajectories as in (c). i, UMAP embeddings of all twelve latent variables for the same experimental trajectories as in (a) and (c), color-coded by dose rate. j, UMAP embeddings of all twelve latent variables for the same experimental trajectories as in (a), (c), and (i), color-coded by non-Gaussianity ($\tau = 1$) on a symmetric logarithmic scale (SymLogNorm) k, UMAP embeddings of all twelve latent variables for the same experimental trajectories as in (a), (c), and (i), color-coded by velocity autocorrelation ($\tau = 1$). l, UMAP embeddings of all twelve latent variables for the same experimental trajectories as in (c) and (g), color-coded by particle size. m, UMAP embeddings of all twelve latent variables for the same experimental trajectories as in (c), (g), and (l), color-coded by non-Gaussianity ($\tau = 1$) on a symmetric logarithmic scale (SymLogNorm) n, UMAP embeddings of all twelve latent variables for the same experimental trajectories as in (c), (g), and (l), color-coded by velocity autocorrelation ($\tau = 1$).

It would also be interesting (although perhaps slightly beyond the scope of this work) to see what latent space is generated if, instead of an experimental LPTM trajectory, LEONARDO is fed a numerically generated anomalous diffusion from the five models considered.

Regarding the UMAP of latent variables when LEONARDO is fed numerically generated trajectories from the five anomalous diffusion models, this is indeed an interesting visualization, and we have added this analysis to the Supplementary Information, and attached it below.

To complement the results reported in Fig.2 the authors should also plot the UMAP representation of the 128-dimensional layer they use to compute the FD. This would help visualise how the generated LPTM trajectories are closer to other LPTM trajectories than to the other five types of anomalous diffusion models.

We agree with the reviewer that this is a valuable visualization, and we have now included it in the Main text Figure 2 panel f as attached below. This UMAP provides an intuitive view of how the different trajectory classes relate to each other in the feature space, and we observe that LPTM and LEONARDO-generated trajectories exhibit significant overlap. We thank the reviewer for this helpful suggestion.

Fig. 2 Evaluation of the model performance of LEONARDO. **a**, Representative trajectories of different diffusion classes, including Brownian Motion (BM), five anomalous diffusion classes (FBM, CTRW, LW, SBM, and ATTM), and experimental trajectories from LPTEM. **b**, Schematic illustrating the generation of trajectories by LEONARDO. Trajectories are sampled from a Gaussian distribution in LEONARDO's latent space and decoded to produce synthetic trajectories. **c**, Schematic of the MoNet2.0 classifier architecture, trained using the trajectories from panel (a). To evaluate the performance of LEONARDO, trajectories from both panels (a) and (b) are input into the trained MoNet2.0 classifier. **d**, Confusion matrix summarizing the classification performance of the MoNet2.0 model on benchmark diffusion classes, demonstrating high accuracy across most classes. **e**, Schematic and results of the Fréchet Distance (FD) calculation. The second-last layer output of MoNet2.0 is used to compute the FD scores between pairs of diffusion class distributions. The lower triangular matrix shows that the FD score between LPTEM and LEONARDO-generated trajectories is significantly lower than the scores between other diffusion classes. **f**, UMAP of the second-last layer of MoNet2.0 showing significant overlap between LPTEM and LEONARDO-generated trajectories. **g**, Classification results of LEONARDO-generated trajectories by MoNet2.0 show that over 95% of LEONARDO-generated trajectories are classified as LPTEM, with smaller fractions classified as ATTM (3.53%) and FBM (0.34%).

Confusion matrix

Looking at Fig. 2, I find it very surprising the Levy walks are identified with 100% accuracy. In my experience, I have never witnessed a 100% accuracy by classifiers. For example, the best method in the AnDi competition identified Levy walks with 97% accuracy in 1D, and the paper describing this method (Argun *et al* 2021 *J. Phys. A: Math. Theor.* **54** 294003), even in the case with the lowest signal-to-noise ratio obtained an accuracy of 98% on 1D trajectories. Perhaps the authors are using the AnDi dataset without measurement noise but even then, it is surprising to achieve a 100% accuracy.

We thank the reviewer for this observation. To clarify, we did not use the AnDi challenge datasets themselves. Instead, we used the AnDi challenge models to generate new 2D trajectories from each class,

which were free of additional noise. This likely explains the relatively high accuracy in classification of Levy Walk trajectories in our case. We have made this clearer in Section 2.2 of the Main text:

“To train MoNet, we used a diverse dataset comprising LPTEM trajectories, Brownian Motion (BM) and five classes of anomalous diffusion processes simulated using the models from the Anomalous Diffusion (AnDi) challenge[Munoz-Gil et al., 2021] without additional noise...”

“Self” Fréchet distance

Would it make sense to report a “self” Fréchet distance, meaning the Fréchet distance between two sets of trajectories of the same class? This would help provide a scale for the magnitude of the FD.

We thank the reviewer for the constructive suggestion regarding the interpretation of the Fréchet distance (FD) values. We agree with the reviewer that providing a comparison to intra-class FD scores is important to help readers better understand and contextualize the significance of the FD score between LEONARDO-generated and experimental LPTEM trajectories.

We have now added a new figure in the Supplementary Information shown below (Figure S4), which shows a lower triangular matrix of FD values computed between two independent batches of each diffusion class. This provides a clear reference for the typical intra-class FD scores, which serve as a baseline for interpreting the FD score between LEONARDO and LPTEM.

In addition, we have added the following clarifying sentence to the Main text:

“The scores computed between two independent batches of the same diffusion class (e.g., FBM vs. FBM, CTRW vs. CTRW, etc.) range from 0.19 to 1.74 (see Figure S4), providing a baseline for interpreting the Fréchet distance values.”

Fig. S4: Fréchet Distance lower triangular matrix showing scores between pairs of diffusion class including intra-class scores. The second-last layer output of MoNet2.0 is used to compute the FD scores between different diffusion classes and between batches of the same class (intra-class scores). The matrix shows that the FD scores between LPTEM and LEONARDO-generated trajectories are significantly lower than the scores between other diffusion comparisons, while intra-class scores, ranging from 0.26 to 1.12, provide a lower bound for contextual comparison.

Connection between latent variables and non-Gaussianity

I find the connection between the hidden variables and the non-Gaussianity shown in Fig.4a and e very weak. Panel a displays a marked difference, while panel e a barely noticeable one. The authors should also discuss why they report the max of the two latent variables as the informative quantity. Also, panel g does not show a difference as striking as the one in panel c.

We thank the reviewer for these observations. To ensure a fair comparison across experimental conditions, we now use 4000-frame trajectories (segmented into 200-frame pieces) for all cases. We have also updated the dose comparison to use doses of 20 vs. 35 instead of 20 vs. 30 (in the previous version), as this comparison between two extreme dose rates provides a clearer distinction between the two extreme conditions. Additionally, we now report the average of μ_1 and μ_5 instead of the max of z_1^2 and z_5^2 ensuring a more direct comparison with non-Gaussianity while reducing sampling noise in the latent space by plotting μ instead of z (as suggested by reviewer 1). Furthermore, we have incorporated UMAP representations of all twelve latent variables color-coded by experimental conditions, non-Gaussianity, and velocity autocorrelation, which further confirm that the latent space encodes statistical properties of the trajectories that are influenced by experimental conditions such as dose and particle size. Please see the revised version of Figure 4 attached above. We thank the reviewer for the helpful suggestions.

MINOR COMMENTS

The authors should define the F1 score.

We have added the following definition of the F1 score in the Methods Section of the Main Text.

“Calculation of F1 Score

The F1 score is a metric that combines precision and recall as follows:

$$F1 = 2 \cdot \frac{\text{Precision} \cdot \text{Recall}}{\text{Precision} + \text{Recall}}$$

Where precision is given by:

$$\text{Precision} = \frac{TP}{TP + FP}$$

And recall is given by:

$$\text{Recall} = \frac{TP}{TP + FN}$$

Here, TP refers to the number of true positives, FP to false positives, and FN to false negatives. The final F1 score for MoNet2.0 was calculated by averaging the F1 scores across all diffusion classes as follows:

$$F1_{\text{weighted}} = \sum_{i=1}^N w_i \cdot F1_i$$

Where N is the total number of classes, $F1_i$ is the F1 score for class i , and w_i is the weight for class i , given by:

$$w_i = \frac{n_i}{\sum_{j=1}^N n_j}$$

Where n_i is the number of trajectories in class i ."

RESPONSE TO REVIEWER COMMENTS

Reviewer #1 (Remarks to the Author):

I thank the authors for responding to my comments and adapting their manuscript based on the raised concerns. I think the manuscript has greatly improved through the revision process and is ready for publication.

I have just some minor comments left, which may not need any further change to the manuscript.

First, I must confess that I don't like the addition of a UMAP, as I think that applying it to the latent space of a properly trained VAE shouldn't give rise to any more information. I also understand that this was added as a response to Reviewer 2.

Let me justify the previous: if the VAE has been properly trained, this means that it has reached a minimal representation of the data, through collapsing all unnecessary neurons to the prior. Hence, a UMAP or any other dimensionality reduction method cannot reduce the dimension any more. They are just going to perform a non-linear transformation of what is already encoded in the surviving neurons. One of the advantages of VAE, in my opinion, is that they avoid the use of such dimensionality reduction methods, and their representations are typically much more interpretable (as shown by this work and many others). As a curiosity, it has been shown that VAEs actually perform PCA in the latent space (see <https://arxiv.org/abs/1812.06775>).

My second minor comment relates to the values of the $\log 2$ and the general training of the VAE. Looking at the table shown in the author's response, I think the training of the VAE can probably be improved. As the authors show throughout the paper, only two parameters are needed to describe the data: velocity autocorrelation and non-Gaussianity. I understand their comment about the network splitting the non-Gaussianity in two dimensions, although in the ideal scenario, I would expect that a single neuron should be sufficient for homogenous diffusion. What concerns me is the high values of $\log 2$, as the authors also note in their response. I believe their high value is due to a too small regularization of the latent space (i.e. too small w_2 weight in their loss). Typically, finding the correct value for such a term is the main challenge in training VAEs, and given that the authors have a quite complex loss, it may be even harder. Perhaps of interest to the authors is the TC-VAE loss (Eq 2 in <https://arxiv.org/pdf/1802.04942>), which typically gives rise to much better disentangled representations in the latent space and is easier to hyperparameterize.

Note that, if one wants to use VAEs in a completely unknown scenario, we need to ensure that the least prior-knowledge is used in the analysis of the latent space, hence the need for models that isolate extremely well all physical variables in only the necessary latent neurons.

Reviewer #1 (Remarks on code availability):

The code provided by the authors has allowed me to test their method and is a valuable tool for researchers reading their paper.

We sincerely thank the reviewer for their thoughtful engagement throughout the review process. We greatly appreciate the recognition that the manuscript has improved and that the provided code is a valuable contribution.

Regarding the addition of UMAPs, we fully agree with the reviewer that a properly trained VAE already encodes the relevant structure in the latent space. Our intent in including UMAP was not to extract new information, but rather to provide a 2D visual aid for readers to intuitively see how the latent space reflects differences in the underlying statistical properties. Since the latent space is 12-dimensional, this projection serves purely as a qualitative illustration.

On the point about the VAE training and the relatively high variance of some latent variables, we appreciate the reviewer's suggestion. As noted previously, our loss function includes multiple statistical terms, and the weights were tuned to strike a balance between encouraging disentangled representations and preserving meaningful reconstruction of statistical properties. In particular, increasing the KL divergence weight led to a sharp degradation in the reconstruction performance across our physics-informed loss terms. We thank the reviewer for pointing us to the TC-VAE formulation and agree that it is an interesting and promising direction to explore in future work.

Reviewer #2 (Remarks to the Author):

The authors have addressed most of my comments.

However, two of my comments were not discussed in sufficient depth and were ignored in the revised version of the manuscript.

The authors did not include any discussion in the manuscript about the relation between the classifiers and the Fréchet distance and their abilities to measure the similarities of the generated trajectories with the experimental dataset (my second comment). The reply to my comment in the rebuttal, though not really exhaustive, highlights some subtleties that the authors glossed over in the manuscript. I therefore invite the authors to expand this discussion and to include at least a summary of it in the revised manuscript.

We thank the reviewer for the suggestion and agree that a clearer explanation of the relationship between the classifiers and Fréchet Distance (FD) would strengthen the manuscript. We have now added a summary discussion at the end of Section 2.2 of the main text to make the required clarification:

“We note that while both the classifier and FD are derived from the same MoNet 2.0 model, they serve different purposes and are not expected to align exactly. The classifier assigns labels based on the most discriminative features in the learned feature space, and therefore highlights whether two trajectories can be separated by a sharp decision boundary. In contrast, the FD measures global statistical similarity between two distributions across the entire 128-dimensional feature space. As a result, two classes may be statistically close in terms of overall feature distributions (i.e., low FD), yet still be confidently separable by the classifier. This is seen, for example, in the case of Lévy Walk and Brownian Motion, which have a relatively low FD but are never confused in the classification task (Figure 2d). Taken together, the FD and classification metrics offer complementary perspectives, and both indicate that LEONARDO-generated trajectories closely resemble the experimental LPTEM trajectories.”

Similarly, in my first comment, I asked to provide additional metrics to quantify the quality of the generated trajectories.

While the FID is an established metric for studying generated images, it has not been widely used for assessing the quality of generated stochastic time series (in case I am wrong, I'd be happy to see references about this). It is therefore not sufficient to blindly take it as the only relevant metric, and additional ones are required. Even though the metrics I

suggested are used for the training, it is not uncommon to show the performance of machine-learning models on a test dataset in terms of the metrics used for training. Especially, highlighting how different such metrics are with respect to those of other anomalous diffusion metrics will strengthen the claims of the authors.

We thank the reviewer for this helpful suggestion. As the reviewer noted, it is very common to evaluate a model's performance on a test dataset using the same metrics used in the loss function during training. We note that Figure S1 of the supplementary indeed reports those quantities, where we reported the values of each loss term, including all statistical metrics, on an independent test set throughout training.

To further strengthen this analysis, we have now added a new subsection in the Main text titled *"Quantitative Comparison of Statistical Properties."* In this section, we compared LEONARDO-generated trajectories (sampled randomly from the latent space) to experimental LPTM trajectories and to six reference diffusion classes: Brownian motion, FBM, CTRW, Lévy Walk, ATTM, and SBM. We computed ensemble-averaged statistical properties over 3,202 trajectories in each case and calculated the squared differences, as reported in Table S1. We found that the LEONARDO-generated trajectories have lower total squared difference with LPTM trajectories than with any of the other diffusion classes. We note that this analysis differs from the loss function used in training, where some metrics are computed per trajectory and others over an ensemble. We then evaluated LEONARDO-reconstructed trajectories, obtained by inputting experimental LPTM trajectories into the trained model and reconstructing them, and found that the reconstructed trajectories yield significantly lower errors than any of the generated comparisons, as expected.

The following section is added to the main text.

"Quantitative Comparison of Statistical Properties"

In addition to the FD and classification-based evaluation metrics, we further evaluated the statistical fidelity of LEONARDO's outputs by comparing both reconstructed and generated trajectories to experimental LPTM trajectories. Specifically, we computed the ensemble-averaged values of each statistical metric used in the physics-informed loss function over 3,202 trajectories (validation dataset size) and calculated the squared differences between these averages. For LEONARDO-generated trajectories sampled randomly from the latent space, we measured how their average statistical properties differ from those of experimental LPTM trajectories. As a reference point, we reported the same difference between the LEONARDO-generated trajectories and six different diffusion classes: Brownian

motion, FBM, CTRW, Lévy Walk, ATTM, and SBM. We then evaluated LEONARDO-reconstructed trajectories, obtained by inputting experimental LPTEM trajectories into the trained model, and compared them statistically to the original LPTEM inputs. As shown in Table S1, LEONARDO-generated trajectories exhibit lower total weighted and unweighted squared errors against LPTEM compared to any reference diffusion class. The comparison between LEONARDO-reconstructed and original LPTEM input data demonstrates that LEONARDO can accurately recover the statistical properties of experimental trajectories, yielding significantly lower weighted and unweighted squared differences compared to reference comparisons between LEONARDO-generated trajectories and experimental LPTEM data, or simulated diffusion classes.”

We have also attached the Table S1 referenced in the Main text below:

Metric	LEONARDO-Generated vs.							LEONARDO-Reconstructed vs.
	LPTEM	Brownian	FBM	CTRW	Lévy Walk	SBM	ATTM	LPTEM
Mean (10^{-7})	5.13	0.25	0.24	0.80	0.20	0.30	1.09	0.27
Variance (10^{-6})	8.30	41.20	26.10	31.80	63.90	40.90	38.40	2.59
Skewness (10^{-2})	1.06	0.03	0.02	0.04	0.07	0.03	0.15	0.05
Kurtosis	0.21	20.27	20.28	4432.95	24.61	5.96	280.20	0.19
Median (10^{-8})	19.58	6.01	8.49	6.31	9.56	5.28	11.00	21.93
Velocity Autocorr (10^{-6})	1.11	21.70	0.77	22.00	713.98	21.80	21.60	0.01
Batch Velocity Autocorr (10^{-6})	2.05	15.00	0.00	24.10	4126.94	19.80	20.70	0.92
XY Correlation (10^{-5})	2.61	23.41	25.50	47.51	30.81	21.83	7.82	0.01
Position Autocorr (10^{-3})	1.04	7.90	4.50	0.08	28.16	4.57	1.34	0.70
Total Weighted Squared Error	2.33	48.03	2.97	312.43	649.39	46.86	63.29	0.14
Total Unweighted Squared Error	0.22	20.28	20.28	4432.95	24.65	5.96	280.20	0.19

Table S1: Statistical comparison of LEONARDO-generated and LEONARDO-reconstructed trajectories against experimental LPTEM trajectories and reference diffusion classes. The first seven columns (LEONARDO-generated) report the squared differences between the average statistical properties of LEONARDO-generated trajectories (sampled randomly from the latent space) and those of experimental LPTEM trajectories, as well as independent batches of Brownian motion, FBM, CTRW, Lévy Walk, SBM, and ATTM. The last column reports the squared differences between the same LPTEM trajectories as the first column, and LEONARDO-reconstructed trajectories, obtained by inputting the LPTEM trajectories into the trained model. This column highlights the high fidelity of LEONARDO’s reconstructions, which yield significantly lower errors for most statistical properties compared to the reference errors observed for LEONARDO-generated trajectories against LPTEM and other diffusion classes. Each metric is computed over an ensemble of 3,202 trajectories. The last two rows summarize the total weighted (using the same weights as in the LEONARDO training loss; see Section 3.1 of the SI) and unweighted squared errors. Lower values indicate higher statistical similarity.

We would like to thank the reviewer for their thoughtful comments and valuable feedback throughout the review process, which significantly strengthened the manuscript.

I highly appreciate the effort that the authors have put in answering my concern. I think the manuscript has greatly improved. I still have a few questions, mostly related to the core ideas behind the work, and what we can actually learn from the trained machine.

Response to Comment 1.1

I thank the authors for their response. I agree with their perspective on the Fernández-Fernández et al. loss, only being valid for Gaussian processes. My comment on the probabilistic loss was indeed more general: given that we are compressing a stochastic signal in the latent space (3 neurons in this case), it is impossible, from an informational perspective, to recover the original input signal, as this was random. It is possible however, to generate trajectories that come from the same distribution. This is what was shown in Fernández-Fernández et al., and what motivates the adequacy of the probabilistic loss vs. an MSE loss, the latter making the training of the network ill-defined by the previous argument.

I believe the rest of the terms in LEONARDO's loss are taking this into account, and the MSE is indeed not needed to reach the results shown in the paper. May this be highlighted by the fact that w_1 in S6 (Supplementary Material) is at least order of magnitude smaller than the rest of the weights?

Response to Comment 1.2

I thank the authors for taking my comments into account about the model's architecture. On this, just a small comment: in Fig. 1b it is not clear that there is a convolutional layer after the attention block of the decoder, as it seems that that attention block directly outputs the trajectory.

About the Fréchet distance calculated, what is the typical value for matching distributions? It may be useful for the reader to know what is the FD at the diagonal (e.g. FBM vs FBM, CTRW vs CTRW,...) to then be able to better compare what the 0.547 of the Generated vs. LPTM means.

Response to Comment 1.3 and 1.4

I sincerely thank the authors for considering my comments and the extensive work they put into answering my questions. I have still few concerns relating this two comments:

- 1) Related to the factors of variation learned by the VAE: the new plots in Fig.4 a+e, b+f,... highlight that there is a connection between the latent neurons and the dose rates and nanorods sizes. Reformulating my comment from the previous review round: what happens when you plot the latent neurons in Fig3c-j w.r.t. to the two experimental parameters? Ultimately, has the model learned to represent in the latent space: a) the size and dose or b) the non-Gaussianity and the velocity autocorrelation? I understand that these two sets are proportional to each other (as shown in Fig. 4), but it would be interesting to know if the authors have any insight on whether the VAE has learned one or the other.
- 2) Minor comment: both in Fig. 3 and 4, the authors show the value z of the latent neuron. However, it is usually sufficient to look at the value μ , as the value of σ is typically independent of the input sample and gives no information about it. It is indeed μ which is encoding the factors of variations. By plotting it instead of z , you usually eliminate unnecessary fluctuations in the plots and can more easily show what is exactly learned. For instance, in Fig4d, the purple plot arises from having multiple μ distributed across the domain or because of the variance σ_4^2 ? Similar for the rest of the plots.

- 3) Why do three neurons survive, rather than two? Is the diffusion inhomogeneous? If not, why is the non-gaussianity for each component separated in the latent space? If the diffusion is homogeneous, I would have expected only two neurons. If it is not, where is this inhomogeneity coming from? I understand that in this case it would be independent of the dose and size, and would help clarify my point 1) above.

Response to Comment 1.5

First, as commented above, the variance is typically almost constant for any input sample. As soon as the VAE “decides” that a latent neuron encodes a factor of variation, it just lowers its value to something close to zero for any input sample.

Given that, I am very surprised by the values of the variances presented by the authors. Typically, the values of the variances of neurons that encode valid information (e.g. z_1 , z_4 and z_5) are orders of magnitude smaller. It is usual to have $\log(\sigma^2) \ll -4$. For instance, let’s consider that LEONARDO get’s a trajectory with a given $C_v = -0.2$. Then, the encoders processes this information and sets $\mu_4 \rightarrow 0$ (from Fig. 3g). Now, because of the big variance, when you sample z_4 the information of the input C_v is completely noised out, as with that variance it's almost impossible to distinguish e.g. $\mu_4 = 0$ and $\mu_4 = 0,5$. This error will be propagated to the decoder, which will then lead to a bad reconstruction loss.

This is even more surprising for neurons z_1 and z_5 , which both have a much bigger variance. This means that no matter what is encoded in that neuron, it is almost completely noised out by such big variances.

Are the numbers presented correct? If so, can the authors comment on this? I suspect that the reconstruction loss should be very high because of this.

The authors have performed extensive work to address the reviewers' comments. I find this second version of the manuscript has markedly improved. However, some of the changes included require additional explanation before I can recommend the manuscript for publication. Below, I will detail the points that need to be discussed.

MAIN COMMENTS

Quality assessment of the generated trajectories

The authors developed two ingenious ways of quantifying how accurately the generated trajectories represent LPTM trajectories. However, their assessments rely on another neural network, which introduces an additional black box making the procedure difficult to scrutinise. Could the author also report simply the "distance" between the generated LPTM trajectories and the original ones in terms of the statistical features they use for their loss function? For example, the Mean squared error of the mean, of the variance of the displacements, of the skewness, etc.

To give a reference, one could compare them with the same metrics for other anomalous diffusion models, such as the five models analysed by the authors in Fig. 2.

Relation between the classifiers and the Fréchet Distance

I find it difficult to draw a precise connection between the conclusions suggested by the classifier and by the Fréchet Distance (FD). From the FD metrics, the LPTM trajectories seem rather close to the CTRW ones. However, the classifier never mistakes an LPTM trajectory for a CTRW one. It does instead mistake it for BM and FBM in about 1% of the cases each. Therefore, these two ways of determining how close trajectories are seem inconsistent.

I am also confused because in their first submission, the authors modelled LPTM as a mixture of FBM and CTRW. This seems at odds

with the fact that the classifier never mistakes an LPTM trajectory for a CTRW one or vice versa.

UMAP

Could the authors present a UMAP representation of the latent variables colour-coded for the different experimental conditions as the ones shown in Fig. 4? Colour-coding in terms of the non-Gaussianity or autocorrelation could also be informative.

It would also be interesting (although perhaps slightly beyond the scope of this work) to see what latent space is generated if, instead of an experimental LPTM trajectory, LEONARDO is fed a numerically generated anomalous diffusion from the five models considered.

To complement the results reported in Fig.2 the authors should also plot the UMAP representation of the 128-dimensional layer they use to compute the FD. This would help visualise how the generated LPTM trajectories are closer to other LPTM trajectories than to the other five types of anomalous diffusion models.

Confusion matrix

Looking at Fig. 2, I find it very surprising the Levy walks are identified with 100% accuracy. In my experience, I have never witnessed a 100% accuracy by classifiers. For example, the best method in the AnDi competition identified Levy walks with 97% accuracy in 1D, and the paper describing this method (Argun *et al* 2021 *J. Phys. A: Math. Theor.* **54** 294003), even in the case with the lowest signal-to-noise ratio obtained an accuracy of 98% on 1D trajectories. Perhaps the authors are using the AnDi dataset without measurement noise but even then, it is surprising to achieve a 100% accuracy.

“Self” Fréchet distance

Would it make sense to report a “self” Fréchet distance, meaning the Fréchet distance between two sets of trajectories of the same class? This would help provide a scale for the magnitude of the FD.

Connection between latent variables and non-Gaussianity

I find the connection between the hidden variables and the non-Gaussianity shown in Fig.4**a** and **e** very weak. Panel **a** displays a marked difference, while panel **e** a barely noticeable one. The authors should also discuss why they report the max of the two latent variables as the informative quantity.

Also, panel **g** does not show a difference as striking as the one in panel **c**.

MINOR COMMENTS

The authors should define the F1 score.